# Design of Experiments
# for Stochastic Contextual Linear Bandits

**Andrea Zanette**[*]
EECS Department
University of California, Berkeley
Berkeley, CA
zanette@berkeley.edu

**Kefan Dong**[*]
Department of Computer Science
Stanford University
Stanford, CA
kefandong@stanford.edu

**Jonathan Lee**[*]
Department of Computer Science
Stanford University
Stanford, CA
jnl@stanford.edu

**Emma Brunskill**
Department of Computer Science
Stanford University
Stanford, CA
ebrun@cs.stanford.edu

## Abstract

In the stochastic linear contextual bandit setting there exist several minimax procedures for exploration with policies that are *reactive* to the data being acquired. In practice, there can be a significant engineering overhead to deploy these algorithms, especially when the dataset is collected in a distributed fashion or when a human in the loop is needed to implement a different policy. Exploring with a single non-reactive policy is beneficial in such cases. Assuming some batch contexts are available, we design a single stochastic policy to collect a good dataset from which a near-optimal policy can be extracted. We present a theoretical analysis as well as numerical experiments on both synthetic and real-world datasets.

## 1 Introduction

Many settings may substantially benefit from data-driven contextualized decision policies that optimize the desired expected outcome. Online machine learning methods like multi-armed bandits and reinforcement learning, that adaptively change interventions in response to outcomes in a closed loop process (see Figure 1a), may not yet be practical for all domains due to the expertise and infrastructure needed. However running an experiment with a fixed decision policy to identify a good personalized policy is likely to be both simple logistically (since currently such decision policies are often specified by hand) and more easily accepted, since many areas (education, healthcare, social sciences) commonly deploy experiments across a few conditions to find the best approach. For example, education startups, political campaigns, and governmental agencies can use email and

---

[*]The first three authors contributed equally. The work was fully completed while Andrea Zanette was a PhD Candidate in the Institute for Computational and Mathematical Engineering at Stanford University. Future revision of this article will be made available at `https://arxiv.org/abs/2107.09912`.

35th Conference on Neural Information Processing Systems (NeurIPS 2021).

text messages to provide targeted information and opportunities, such as information to encourage vaccination, or tips to parents to support their child's developmental stage. Such organizations are familiar with standard experimental design, and but generally lack the infrastructure for continuous online contextual MAB learning. Experiments that involve deploying a nonadaptive policy that fit in with standard workflows but enable data-efficient learning of contextualized policies might offer substantial benefits over AB testing or relying on other segmentation methods that may not directly optimize desired outcomes.

For these reasons, a key opportunity is to design *static* or *nonadaptive* policies that can be used to gather data to identify optimal contextualized decision policies. Indeed a fixed data collection strategy is practically desirable (1) whenever multiple agents collect data asynchronously and communication to update their policy is difficult or impossible, and (2) whenever changing the policy requires a significant overhead either in the engineering infrastructure or in the training of human personnel. Several prior papers limit the number of policy switches with minimal sample complexity impact (Han et al., 2020; Ruan et al., 2020; Ren et al., 2020; Bai et al., 2019; Wang et al., 2021). Motivated by the above settings, we look for a *single, nonadaptive* policy for data collection.

**Setting and goal** We consider the linear stochastic contextual bandit setting where each context $s \in \mathcal{S}$ is sampled from a distribution $\mu$ and a context-dependent action set $\mathcal{A}_s$ is made available to the learner. The bandit instance is defined by a feature extractor $\phi(s, a) \in \mathbb{R}^d$ and some unknown parameter $\theta^\star \in \mathbb{R}^d$. Upon choosing an action $a \in \mathcal{A}_s$, the linear reward function $r(s, a) = \phi(s, a)^\top \theta^\star + \eta$ is revealed to the learner corrupted by mean zero 1-subGaussian noise $\eta$. Our goal is to construct an exploration policy $\pi_e$ to gather a dataset, such that after that dataset is gathered, we can extract a near optimal policy $\hat{\pi}$ from it. In particular, we want to minimize the number of exploration samples.

Perhaps surprisingly, there has been relatively little work on this setting. Prior work on exploration to quickly identify a near-optimal policy focuses on best-arm identification using adaptive policies that react to the observed rewards (Soare et al., 2014; Tao et al., 2018; Jedra and Proutiere, 2020) or design of experiments that produces a nonadaptive policy for data collection (Kiefer and Wolfowitz, 1960; Esfandiari et al., 2019; Lattimore and Szepesvari, 2020); both lines of work assume that a *single*, repeated context with unchanging action set is presented to the learner. In contrast we are interested in identifying near-optimal context-specific decision policies. The closest related work (Ruan et al., 2020) investigates our task as a subtask for online regret learning, but requires an amount of data that scales as $\Omega(d^{16})$, which is impractical in applications with even moderate $d$.[2]

Without any apriori information, no algorithm can do much better than deploying a random policy, which can require an amount of data that scales exponentially in $d$, see appendix G.1. However, in many common applications, *prior data about the context distribution $\mu(s)$* and the state–action feature representation $\phi$ is available. For example, an organization will often know information about its customers and specify the feature representation used for state–action spaces in advance.[3] The initially available state contexts are referred to as *offline* (state) contexts data.

Our algorithm leverages historical context data to enable data efficient design of experiments for stochastic linear contextual bandits. It uses offline context-only data $\mathcal{C}$ to design a nonadaptive policy $\pi_e$ to collect new, *online* data where reward feedback is observed (see Figure 1b), and uses the resulting dataset $\mathcal{D}'$ to learn a near-optimal on average decision policy $\hat{\pi}$ for future use. We highlight that the algorithm does not get to adjust the exploratory policy $\pi_e$ while the online data is being collected.

**Contributions** We make the following contributions.

- Using past state contexts only, we design a single, nonadaptive policy to acquire online data that can be used to compute a context-dependent decision policy that is near-optimal in expectation across the contexts with high probability, for future use.
- To identify an $\epsilon$-optimal policy, our algorithm achieves the minimax lower bound $\Omega(\min\{d \log \sum_s \mathcal{A}_s, d^2)\}/\epsilon^2)$ on the number of online samples (ignoring log and constants), while keeping the number of offline state contexts required polynomially small ($O(d^2/\epsilon^2)$ or $O(d^3/\epsilon^2)$).

---

[2]Note that the number of offline data in (Ruan et al., 2020) is independent of $1/\epsilon$. But for most of the practical settings, our bound $d^3/\epsilon^2$ can be much smaller than $d^{16}$.

[3]In other words, given a set of previously observed states $s_1, \ldots, s_M$, and a known state–action representation $\phi$, for any potential action $a$, we can compute the resulting representation $\phi(s, a)$.

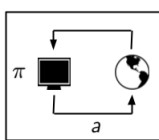 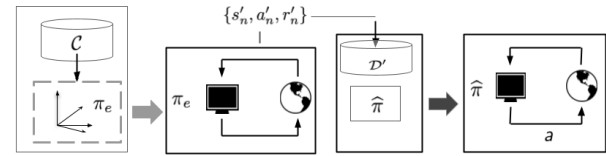

(a) Traditional online RL framework          (b) Design of experiment with an online and offline component

Figure 1: Comparison between the traditional RL setting and design of experiments

- Our experiments on a simulation and on a learning to rank Yahoo dataset show our strategic design of experiments approach can learn better decision policies with less exploration data compared to standard static exploration strategies that fail to exploit structure.

## 2   Related Work

Most linear bandit papers focus on online regret. Research on sample complexity results has focused primarily on adaptive algorithms for non-contextual bandits with a simple or vectored action space (Soare et al., 2014; Tao et al., 2018; Xu et al., 2018; Fiez et al., 2019; Jedra and Proutiere, 2020; Degenne et al., 2020). Their sample complexity bounds depend on the suboptimality gap of the action set, and are therefore, instance-dependent. The design of experiments literature (Kiefer and Wolfowitz, 1960; Esfandiari et al., 2019; Lattimore and Szepesvari, 2020; Ruan et al., 2020) has also focused on non-contextual bandits, but in designing a non-adaptive policy such that the resulting dataset can be used to identify an optimal arm with high probability.

A few recent papers consider learning the optimal policy for contextual bandits in the pure exploration setting (Deshmukh et al., 2018; Ruan et al., 2020). Deshmukh et al. (2018) propose an algorithm that yields an online reactive policy for when the reward is generated by a RKHS function on the input space: in contrast we focus on nonadaptive policies for linear contextual bandits.

There has been recent interest in reward-free reinforcement learning in Markov decision processes (e.g. (Jin et al., 2020; Wang et al., 2020a)). This work is focused on online adaptive policies. Jin et al. (2020)'s algorithm is similar in structure to our's and could likely be adapted to the offline, bandit setting, but they focus on tabular domains. Note for simple contextual multi-armed bandits (tabular states, no shared structure, Jin et al.'s setting with H=1), a good strategy in the non-adaptive setting is to simply sample actions uniformly; indeed this is optimal given the lack of shared information between state-actions. No sophisticated algorithm is needed in that case.

For linear bandits, the closest reward-free work we are aware of is work on RL with linear function approximationWang et al. (2020a). While their algorithm can be specialized to the linear bandit setting and modified to match our algorithm by using offline data, the theoretical analysis would need to be very different from that in(Wang et al., 2020b)'s paper (and thus it would not yield the results presented in our paper). A key challenge in our case is to compute a non-adaptive policy for gathering data and be able to ensure the covariance matrix of the resulting data will be very similar to the covariance matrix that could result by a simulated adaptive policy. This issue does not arise in (Wang et al., 2020b) because they focus on reactive policies.[4]

The most related paper to this work is by Ruan et al. (2020). The main focus of Ruan et al. (2020) is to minimize the number of policy switches in the online phase while keeping the expected regret optimal. Directly applying Theorem 6 of Ruan et al. (2020) yields a $\widetilde{O}(d^2/\epsilon^2)$ online sample complexity and $\widetilde{O}(d^{16})$ offline sample complexity. Their offline sample complexity is independent of $\epsilon$. As a trade-off, Ruan et al. (2020) have a highly suboptimal dependence on $d$, which leads to burn-in phase too large to be practically useful; their algorithm also suffers from a high computational complexity. Note their work is focused on a different objective than our's.

---

[4]In particular, we will shortly introduce a planner-sampler two stage algorithm. We decouple the planner from the sampler to identify a non-reactive policy. Due to the decoupling, the planner needs to accurately predict what will happen when the sampler is run. To ensure this, a key challenge is to ensure that the planner inverse covariance matrix is very similar to the sampler inverse covariance matrix.

# 3   Setup

We consider the stochastic linear contextual bandit model with stochastic contexts. A bandit instance is characterized by a tuple $\langle \mathcal{S}, \mathcal{A}, \mu, r \rangle$ where $\mathcal{S}$ is the context space and $\mu$ is the context distribution. For a context $s \in \mathcal{S}$, the action space is denoted by $\mathcal{A}_s$. The feature map $\phi : (s, a) \mapsto \phi(s, a) \in \mathbb{R}^d$ is assumed to be known and defines the linear reward model $r(s, a) = \phi(s, a)^\top \theta^\star + \eta$ for some $\theta^\star \in \mathbb{R}^d$ parameter and some mean-zero 1-subgaussian random variable $\eta$.

We occasionally use the $\widetilde{O}$ notation to suppress $\mathrm{polylog}$ factors of the input parameters $d, \lambda, \frac{1}{\delta}$. We write $f \lesssim g$ if $f = O(g)$, and $f \lesssim\!\!\!\sim g$ if $f = \widetilde{O}(g)$. We say $f = \Omega(g)$ if there exists a positive constant $c$ such that $f \geq cg$. For a positive semi-definite matrix $\Sigma \in \mathbb{R}^{d \times d}$ and a vector $x \in \mathbb{R}^d$, let $\|x\|_\Sigma = \sqrt{x^\top \Sigma x}$. For two symmetric matrices $A, B$, we say $A \preceq B$ if $B - A$ is positive semi-definite.

Our analysis consists of an offline and online component. We often add $'$ to indicate the online quantities, e.g., we denote with $s_1, s_2, \ldots$ the offline contexts and with $s'_1, s'_2, \ldots$ the online contexts.

## 3.1   Objective and Error Decomposition

As depicted in Figure 1, our approach is to leverage offline state contexts $\mathcal{C} = \{s_1, \ldots, s_M\}$ where $s_m \sim \mu$ to design a single stochastic policy $\pi_e$ to acquire data from an online data stream $\mathcal{C}' = \{s'_1, \ldots, s'_N\}$. This generates a dataset $\mathcal{D}' = \{(s'_n, a'_n, r'_n)\}_{n=1,\ldots,N}$ where $s'_n \sim \mu, a'_n \sim \pi_e(s'_n)$ and $r'_n = r(s'_n, a'_n)$. Using $\mathcal{D}'$, the least-square predictor $\widehat{\theta}$ and the corresponding greedy policy $\widehat{\pi}$ can be extracted

$$\widehat{\theta} = \left(\Sigma'_N\right)^{-1} \sum_{i=1}^N \phi(s'_n, a'_n) r'_n, \qquad \widehat{\pi}(s) = \arg\max_{a \in \mathcal{A}_s} \phi(s, a)^\top \widehat{\theta} \tag{1}$$

where $\Sigma'_N = \left( \sum_{n=1}^N \phi(s'_n, a'_n) \phi(s'_n, a'_n)^\top + \lambda_{reg} I \right)$ is the empirical cumulative covariance matrix with regularization level $\lambda_{reg}$. The quality of the dataset $\mathcal{D}'$ (and of the whole two-step procedure) is measured by the suboptimality of the extracted policy $\widehat{\pi}$ obtained after data collection:

$$\mathbb{E}_{s \sim \mu}[\max_a \phi(s, a)^\top \theta^\star - \phi(s, \widehat{\pi}(s))^\top \theta^\star]. \tag{2}$$

Note that Eq. (2) measures the expectation over the contexts of the suboptimality between the resulting decision policy $\widehat{\pi}$ and the optimal policy. This is a looser criteria than a maximum norm bound which evaluates the error over any possible context $s$: in general this latter error may not be easily reduced if certain directions in feature space are rarely available.

A related objective is to minimize the maximum prediction error on the linear bandit instance. By least square regression analyses (Lattimore and Szepesvári, 2020), with probability at least $1 - \delta$ we have

$$\mathbb{E}_{s \sim \mu} \max_a |\phi(s, a)^\top (\theta^\star - \widehat{\theta})| \leq \sqrt{\beta} \, \mathbb{E}_{s \sim \mu} \max_a \|\phi(s, a)\|_{(\Sigma'_N)^{-1}}, \tag{3}$$

where

$$\sqrt{\beta} = \min \left\{ \underbrace{\sqrt{2 \ln 2 |\sum_s \mathcal{A}_s| + \ln \frac{1}{\delta}}}_{\text{Small } \sum_s \mathcal{A}_s}, \underbrace{2\sqrt{2d \ln 6 + \ln \frac{1}{\delta}}}_{\text{Large } \sum_s \mathcal{A}_s} \right\} + \underbrace{\sqrt{\lambda_{reg}} \|\theta^\star\|_2}_{\text{Regularization Effect}}. \tag{4}$$

The above expression assumes that the state-action-rewards $(s'_n, a'_n, r'_n)$ are drawn i.i.d. from a fixed distribution. This is satisfied in our setting as the data-collection policy $\pi_e$ is nonadaptive to the online data. The parameter $\beta$ governs the sample complexity as a function of the size of the state-action space and also highlights the impact of the regularization bias.

Small predictive error (Eq. (3)) can be used to bound the suboptimal gap of the greedy policy (Eq. (2)). Therefore to obtain good performance, it is sufficient to bound $\mathbb{E}_{s \sim \mu} \max_a \|\phi(s, a)\|_{(\Sigma'_N)^{-1}}$. This can be achieved by designing an appropriate sampling policy $\pi_e$ to yield a set of $(s'_n, a'_n, r'_n)$ tuples whose empirical cumulative covariance matrix $\Sigma'_N$ is as 'large' as possible.

# 4 Algorithms

**Reward-free LINUCB** First, assume it is acceptable to have an algorithm that updates its policy reactively. In order to reduce $\mathbb{E}_{s\sim\mu}\max_a\|\phi(s,a)\|_{(\Sigma'_n)^{-1}}$ we could use an algorithm that, every time a context $s\sim\mu$ is observed, chooses the action $\arg\max_{a\in\mathcal{A}_s}\|\phi(s,a)\|_{(\Sigma'_n)^{-1}}$ where the norm $\|\phi(s,a)\|_{(\Sigma'_n)^{-1}}$ that represents the uncertainty is highest (cf. section 3.1). Such procedure is related (although not identical) to the well known globally optimal design of experiment applied to linear bandits (i.e., Lattimore and Szepesvari (2020)). Algorithmically, it corresponds to running the LIN-UCB algorithm (Abbasi-Yadkori et al., 2011) with the empirical reward function set to zero. In other words, the planner is essentially an *offline, reward-free* version of LINUCB. We emphasize that the associated non-stationary policy is generated offline and is not actively played. One can show that after $\approx d^2/\epsilon^2$ iterations the uncertainty $\sqrt{\beta}\,\mathbb{E}_{s\sim\mu}\max_a\|\phi(s,a)\|_{(\Sigma'_n)^{-1}}\le\epsilon$; if the algorithm stores the observed reward $r(s,a)$ in every visited context $s$ and chosen action $a$, the greedy policy that can be extracted from this dataset (of size $\approx d^2/\epsilon^2$) is $\epsilon$-optimal (cf. Eqs. (2),(3)), as desired.

Unfortunately we cannot run this reward-free algorithm online, as its policy is *reactive* to the online stream of observed online contexts $s$ and selected actions $a$, while we want a nonadaptive policy.

---

**Algorithm 1** PLANNER (Reward-free LINUCB)

1: **Input**: Contexts $\mathcal{C}=\{s_1,\ldots,s_M\}$, reg. $\lambda_{reg}$
2: $\Sigma_1=\lambda_{reg}I$
3: $\underline{m}=1$
4: **for** $m=1,2,\ldots M$ **do**
5:    **if** $\det(\Sigma_m)>2\det(\Sigma_{\underline{m}})$ or $m=1$ **then**
6:       $\underline{m}\leftarrow m$
7:       $\Sigma_{\underline{m}}\leftarrow\Sigma_m$
8:    **end if**
9:    Define $\pi_m:s\mapsto\arg\max_{a\in\mathcal{A}_s}\|\phi(s,a)\|_{\Sigma_{\underline{m}}^{-1}}$
10:   $\Sigma_{m+1}=\Sigma_m+\alpha\phi_m\phi_m^\top;\ \phi_m=\phi(s_m,\pi_m(s_m))$
11: **end for**
12: **return** policy mixture $\pi_e$ of $\{\pi_1,\ldots,\pi_M\}$

**Algorithm 2** SAMPLER

1: **Input**: $\pi_e=\{\pi_1,\ldots,\pi_M\}$, reg. $\lambda_{reg}$
2: Set $\mathcal{D}'=\emptyset$
3: **for** $n=1,2,\ldots N$ **do**
4:    Receive context $s'_n\sim\mu$
5:    Sample $m\in[M]$ uniformly at random
6:    Select action $a'_n=\pi_m(s'_n)$
7:    Receive feedback reward $r'_n$
8:    Store feedback $\mathcal{D}'=\mathcal{D}'\cup\{s'_n,a'_n,r'_n\}$
9: **end for**
10: **return** dataset $\mathcal{D}'$

---

Our algorithm leverages this idea and consists of two subroutines: 1) the *planner* (Alg. 1) which operates on offline contexts and identifies a mixture policy $\pi_e$ (this is the exploratory policy $\pi_e$ mentioned in the introduction) 2) the *sampler* (Alg. 2) which runs $\pi_e$ online to finally gather the dataset. This way, $\pi_e$ is nonadaptive to the online data.

**Planner** The purpose of the planner is to use past contexts to compute the exploratory policy. The planner runs the reward-free version of LINUCB on the offline context dataset $\mathcal{C}$ as described earlier in this section. This way, the planner selects the action $a$ in the current offline context $s_m$ that maximizes the uncertainty encoded in $\|\phi(s_m,a)\|_{\Sigma_{\underline{m}}^{-1}}$ where $\Sigma_{\underline{m}}$ is a scaled, regularized, cumulative covariance over the contexts parsed so far and the actions selected. Note this procedure is possible since the state–action function $\phi(s_m,a)$ is assumed to be known for any input $(s_m,a)$ pair, and no actual rewards are observed. The doubling schedule yields a short descriptor for the planner's policies $\{\pi_1,\ldots,\pi_M\}$. The variable $\underline{m}$ indicates the last doubling update before iteration $m$.

A key choice is the parameter $\alpha\in(0,1]$ in the cumulative covariance matrix update. The rationale is that when $\alpha<1$ each rank-one update $\phi_m\phi_m^\top$ to the cumulative covariance gets *discounted*. The smaller $\alpha$ is, the more offline samples the planner needs to get to a sufficiently positive definite covariance matrix $\Sigma_M$. This choice effectively averages the updates and increases the estimation accuracy of the planner's covariance with respect to its conditional expectation.

**Sampler** Upon termination, the planner identifies a sequence of policies $\pi_1,\ldots,\pi_M$. Now consider the average policy $\pi_e$: every time an action is needed, $\pi_e$ samples one index $m\in[M]$ uniformly at random and plays $\pi_m$. This is the policy that the sampler (Alg. 2) implements for $N=\alpha M\le M$ fresh online contexts. Intuitively, executing the mixture policy $\pi_e$ in the online phase produces the same covariance matrix as $\Sigma_M$ (in expectation).

| Sample Complexity Bounds | | | | | |
|---|---|---|---|---|---|
| | | Small $\sum_s \mathcal{A}_s$ | Large $\sum_s \mathcal{A}_s$ | Large $\sum_s \mathcal{A}_s$ |
| Offline Data | $M$ | $d^3/\epsilon^2$ | $d^3/\epsilon^2$ | $d^2/\epsilon^2$ |
| Online Data | $N$ | $(d \ln \sum_s |\mathcal{A}_s|)/\epsilon^2$ | $d^2/\epsilon^2$ | $d^2/\epsilon^2$ |
| Regularization | $\lambda_{reg}$ | 1 | 1 | $d$ |

Table 1: Sample complexity bounds (ignoring constants and log terms) to obtain an $\epsilon$-optimal policy

Upon playing $\pi_m$ in state $s'_n$, the corresponding reward is observed and the tuple $(\{s'_n, a'_n, r'_n\})$ is stored. Since $\pi_e$ is the average policy played by the planner, we expect that running $\pi_e$ on the online dataset produces a covariance matrix $\Sigma'_N$ similar to the planner's $\Sigma_M$. In a sense, the covariance matrix is a proxy for the amount of information acquired by the algorithm. This means the sampler acquires the same information that the planner would have acquired if it was acting online. Since the planner is the reward-free LINUCB algorithm, we expect its policy to efficiently reduce our uncertainty over the reward parameters and learn a near optimal policy; our analysis will make this intuition precise. Note the sampler's policy is nonadaptive to the online stream of data.

Our work uses mixture policies; mixture policies are not essential, although they arise naturally from the way the algorithms are formulated.

## 5 Main Result

**Theorem 1.** *Fix $\epsilon > 0$, and consider running Alg. 1 for $M = \widetilde{\Omega}(\frac{d^2 \beta}{\lambda_{reg} \epsilon^2})$ iterations and Alg. 2 for $N = \widetilde{\Omega}(\frac{d\beta}{\epsilon^2})$ iterations with regularization $\lambda_{reg} \in (\Omega(\ln(d/\delta)), d]$. Let $\widehat{\theta}, \widehat{\pi}$ be as in Eq.* (1). *For any $\epsilon \leq 1$, with probability at least $1 - \delta$ the expected maximum uncertainty*

$$\mathbb{E}_{s \sim \mu} \max_{a \in \mathcal{A}_s} |\phi(s, a)^\top (\theta^\star - \widehat{\theta})| \leq \epsilon \tag{5}$$

*and the suboptimality of the greedy policy $\widehat{\pi}$ satisfies*

$$\mathbb{E}_{s \sim \mu} \max_{a \in \mathcal{A}_s} (\phi(s, a) - \phi(s, \widehat{\pi}(s)))^\top \theta^\star \leq 2\epsilon. \tag{6}$$

In Table 1 we instantiate the bounds from Theorem 1 in different settings, ignoring constants and log terms. This highlights different tradeoffs between the regularization $\lambda_{reg}$ and the number of offline contexts ($M = N/\alpha$) needed to achieve the online minimax sample complexity lower bound $\approx \min\{d, \ln \sum_s |\mathcal{A}_s|\} \times d/\epsilon^2$ (Chu et al., 2011; Abbasi-Yadkori et al., 2011). [5] In the large action regime $\ln |\sum_s \mathcal{A}_s| \gtrsim d$, a regularization level $\lambda_{reg} \approx d$ gives the optimal online sample complexity $N \approx d^2/\epsilon^2$ while requiring a context dataset of the same size ($M \approx d^2/\epsilon^2$, by choosing $\alpha = 1$).

More often, a lower level of regularization can be preferable to introduce less bias. For example $\lambda_{reg} = 1$ is a common choice in linear bandits (Abbasi-Yadkori et al., 2011; Chu et al., 2011). When the cumulative covariance matrix is less regularized, we may need to compensate with additional offline data to ensure the covariance matrices are accurately estimated, which is achieved by setting $M = N/\alpha \approx dN$ (i.e., $\alpha \approx 1/d$). In this way, additional samples are collected by the planner to maintain its covariance estimation accuracy (and hence its planning accuracy) despite the lower regularization.

Note that our algorithm achieves optimal sample complexity for the online data. Even with adaptive policies, the best known minimax upper bound is $d^2/\epsilon$ (Abbasi-Yadkori et al., 2011; Chu et al., 2011). Although Chu et al. (2011); Abbasi-Yadkori et al. (2011) consider regret minimization instead of sample complexity. But their results can be easily translated to sample complexity bounds (See Lemma 3 in Appendix A.2, and also Jin et al. (2018, Section 3.1)). However, the optimal rate of the number of offline data is unclear. We believe our results can be improved by new tools in matrix concentration, and we leave this direction for future work. Interestingly, in our experiments the algorithm performed well numerically even when $\lambda_{reg} < 1$ and $\alpha = 1$.

---

[5]The minimax lower bounds in the literature are stated for the regret setting, but can nonetheless be adapted to derive sample complexity results. See Appendix A.1 for detailed discussion.

More generally, the amount of regularization should be set following the classical bias-variance tradeoff; its choice is mostly a statistical learning question, and its optimal value is normally problem dependent. As a result, the problem dependent tradeoff between different values of $\lambda_{reg}$ is not reflected in our minimax analyses but can be appreciated in our numerical experiments on the real world dataset.

# 6 Proof Sketch

We give a brief proof sketch where we ignore constants and log factors; the full analysis can be found in the appendix. First we introduce some notation. We define the scaled cumulative covariance $\Sigma_M$ at step $m$ for the planner and the cumulative covariance $\Sigma'_n$ for the sampler at step $n$:

$$\Sigma_m = \alpha \sum_{j=1}^{m-1} \phi(s_j, a_j)\phi(s_j, a_j)^\top + \lambda_{reg} I, \qquad \Sigma'_n = \sum_{j=1}^{n-1} \phi(s'_j, a'_j)\phi(s'_j, a'_j)^\top + \lambda_{reg} I \quad (7)$$

where the planner's $j$ actions is $a_j = \pi_j(s_j)$ and likewise the sampler's $j$ actions is $a'_j \sim \pi_e(s'_j)$.

**Uncertainty Stochastic Process** We represent the value of the uncertainty though a stochastic process. Let us define the filtration for the planner $\mathcal{F}_m = \sigma(s_1, \ldots, s_{m-1})$ at stage $m$ and for the sampler $\mathcal{F}'_n = \sigma(s_1, \ldots, s_M, s'_1, \ldots, s'_{n-1})$ at stage $n$ to represent the amount of information available. Let us also define the observed uncertainty $U_m$ for the planner and likewise $U'_n$ for the sampler:

$$U_m \stackrel{def}{=} \max_{a \in \mathcal{A}_{s_m}} \|\phi(s_m, a)\|_{(\Sigma_m)^{-1}}, \qquad U'_n \stackrel{def}{=} \max_{a \in \mathcal{A}_{s_n}} \|\phi(s'_n, a)\|_{(\Sigma'_n)^{-1}}. \quad (8)$$

We ultimately want to bound $\mathbb{E}'_n U'_n = \mathbb{E}[U'_n \mid \mathcal{F}_n] = \mathbb{E}_{s \sim \mu} \max_a \|\phi(s, a)\|_{(\Sigma'_n)^{-1}}$ when $n = N$, i.e., the average uncertainty at the end, see Section 3. Likewise, let us define $\mathbb{E}_m U_m = \mathbb{E}[U_m \mid \mathcal{F}_m] = \mathbb{E}_{s \sim \mu} \max_a \|\phi(s, a)\|_{(\Sigma_m)^{-1}}$.

We show that $\mathbb{E}'_N U'_N$ is minimized in two steps: 1) we show that the planner's uncertainty $\mathbb{E}_M U_M$ can be bounded 2) since the sampler implements the planner's average policy, $\mathbb{E}'_N U'_N$ cannot be too far from $\mathbb{E}_M U_M$.

We highlight that $M \geq N$ in general, i.e., the stochastic processes proceed at different speed. In fact, the planner needs more data than the sampler ($M \geq N$) as the planner's policy is reactive to the offline contexts; this forces us to use more data-intensive concentration inequalities for the planner.

## 6.1 Offline Uncertainty

The next lemma is formally presented in Lemma 5 and Lemma 7 in the appendix and examines the reduction in the planner's expected uncertainty $\mathbb{E}_M U_M$.

**Lemma 1** (Offline Expected Uncertainty). *With high probability we have*

$$\mathbb{E}_M U_M \leq \frac{1}{M} \sum_{m=1}^{M} \mathbb{E}_m U_m \lesssim \frac{1}{M} \sum_{m=1}^{M} U_m \lesssim \sqrt{\frac{d}{\alpha M}}.$$

*Proof sketch.* We start by modifing a result from the classical bandit literature (e.g., Abbasi-Yadkori et al. (2011)) to bound the average realized uncertainty $U_m$ while accounting for the extra $\alpha$ scaling factor contained in the covariance matrix $\Sigma_M$:

$$\frac{1}{M} \sum_{m=1}^{M} U_m = \frac{1}{M} \sum_{m=1}^{M} \|\phi(s_m, a_m)\|_{\Sigma_m^{-1}} \lesssim \sqrt{\frac{d}{\alpha M}}.$$

The parameter $\alpha$ on the right hand side above arises from its inclusion in the definition of cumulative covariance Eq. 7. This justifies the last inequality in the lemma's statement.

While we bounded $\frac{1}{M} \sum_{m=1}^{M} U_m$, we need to bound the average conditional expectations $\frac{1}{M} \sum_{m=1}^{M} \mathbb{E}_m U_m$. Bernstein's inequality would bound the actual deviation $\sum_{m=1}^{M} U_m$ by the conditional variances (or expectations $\sum_{m=1}^{M} \mathbb{E}_m U_m$ given our boundness assumptions); here we need the opposite, and so we 'reverse' the inequality in Theorem 3 *(Reverse Bernstein for Martingales)* to claim $\sum_{m=1}^{M} \mathbb{E}_m U_m \lesssim \sum_{m=1}^{M} U_m$ with high probability to conclude. □

## 6.2 Online Uncertainty

In the prior section we showed that the planner would be successful in reducing its final uncertainty $\mathbb{E}_M U_M$ if it was collecting reward information; moreover, its sequence of policies $\pi_1, \ldots, \pi_M$ only depends on the observed contexts. Because of this, if the sampler runs the planner's average policy $\pi_e$, we would expect a similar reduction in the uncertainty $\mathbb{E}'_N U'_N$ after $N = \alpha M$ iterations (the $\alpha$ factor simply arises because the planner's information are discounted by $\alpha$). The argument is formalized in Lemma 4 *(Relations between Offline and Online Uncertainty)* in the appendix, which we preview here. We let $K$ be the number of policy switches by the sampler which is bounded by $K \lesssim d$ in Lemma 16 *(Number of Switches)* in appendix.

**Lemma 2** (Relations between Offline and Online Uncertainty). *If* $\lambda_{reg} = \Omega(\ln \frac{d}{\delta})$ *and* $M = \Omega\left(\frac{KN}{\lambda_{reg}} \ln \frac{dNK}{\lambda_{reg}\delta}\right)$, *upon termination of Alg. 1 and 2 it holds with high probability that*

$$\mathbb{E}'_N U'_N \lesssim \mathbb{E}_M U_M.$$

*Proof.* Let $d_1, \ldots, d_M$ be the conditional distributions of the feature vectors sampled at timesteps $1, \ldots, M$ in Alg. 1 after the algorithm has terminated. Conditioned on $\mathcal{F}_M$, the $d_i$'s are non-random. Then the conditional expected covariance matrix given $\mathcal{F}_M$ can be defined as

$$\overline{\Sigma} = \alpha \sum_{i=1}^{M} \mathbb{E}_{\phi \sim d_i} \phi\phi^\top + \lambda_{reg} I \tag{9}$$

Now, our argument relies on some matrix concentration inequalities which hold if

$$\lambda_{reg} = \Omega(\ln \frac{d}{\delta}), \qquad \frac{1}{\alpha} = \Omega\left(\frac{K}{\lambda_{reg}} \ln \frac{dNK}{\lambda_{reg}\delta}\right). \tag{10}$$

Precisely, from Lemma 14 *(Matrix Upper Bound Offline Phase)* and Lemma 15 *(Matrix Upper Bound Online Phase)* with high probability we have that

$$\forall x, \|x\|_2 \leq 1: \qquad \|x\|_{(\Sigma'_N)^{-1}} \lesssim \|x\|_{\overline{\Sigma}^{-1}} \qquad \text{and} \qquad \|x\|_{\overline{\Sigma}^{-1}} \lesssim \|x\|_{\Sigma_M^{-1}}. \tag{11}$$

In words, we can relate the planner's (random) scaled covariance $\Sigma_M$ to its conditional expectation $\overline{\Sigma}$, with the guarantee that it won't be very different from the sampler's covariance $\Sigma'_N$ (which implements the planner's policy). These concentration inequalities are key to the proof and are proved in the appendix.

Define the policy maximizing the online uncertainty $\pi'_n(s) = \arg\max_{a \in \mathcal{A}_s} \|\phi(s, a)\|_{(\Sigma'_n)^{-1}}$. Under the event in Eq. (11) we can write

$$\mathbb{E}'_N U'_N \overset{def}{=} \mathbb{E}_{s \sim \mu} \max_{a \in \mathcal{A}_s} \|\phi(s, a)\|_{(\Sigma'_N)^{-1}} = \mathbb{E}_{s \sim \mu} \|\phi(s, \pi'_N(s))\|_{(\Sigma'_N)^{-1}} \lesssim \mathbb{E}_{s \sim \mu} \|\phi(s, \pi'_N(s))\|_{\overline{\Sigma}^{-1}}$$

$$\lesssim \mathbb{E}_{s \sim \mu} \|\phi(s, \pi'_N(s))\|_{\Sigma_M^{-1}} \leq \mathbb{E}_{s \sim \mu} \max_a \|\phi(s, a)\|_{\Sigma_M^{-1}} = \mathbb{E}_M U_M.$$

$\square$

## 6.3 Conclusion and Tradeoffs

We can now tune the parameters $\alpha, \lambda_{reg}$. Starting from Eq. (3), we can write

$$\mathbb{E}_{s \sim \mu} \max_a \phi(s, a)^\top (\theta^\star - \widehat{\theta}) \leq \sqrt{\beta} \, \mathbb{E}'_N U'_N \lesssim \sqrt{\beta} \, \mathbb{E}_M U_M \lesssim \sqrt{\frac{\beta d}{\alpha M}} = \sqrt{\frac{\beta d}{N}}. \tag{12}$$

Thus, the rate-optimal online sample complexity $N \approx \frac{\beta d}{\epsilon^2}$ always suffices. However, we need a different number of offline contexts depending on the regularization that we want to adopt. Since $K \lesssim d$, the reader can verify that if $\lambda_{reg} \approx d$ then the preconditions in Eq. (10) are verified with $M \approx N$, so $\alpha \approx 1$. If a lower value for the regularization $\lambda \approx 1$ is desired, $M \approx dN$ would be required (thus $\alpha \approx \frac{1}{d}$).

The reason why we might need $M \geq N$ (i.e., $\alpha \leq 1$) is that the planner's policy is adaptive to the observed offline contexts. This means that the planner's cumulative covariance $\Sigma_M$ is not a sum of i.i.d. rank-one random matrices; in this case, we need either a more data-hungry concentration inequality or heavier regularization, leading to the the precondtion on $\alpha$ in Eq. (10).

# 7 Experiments

We now study the empirical properties of the planner and sampler in a synthetic setting and a real-world dataset from the Yahoo! Learning to Rank challenge (Chapelle and Chang, 2011). We also analyze its sensitivity to regularization. Additional details are in the appendix.

## 7.1 Synthetic setting

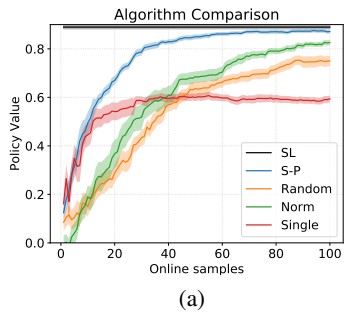

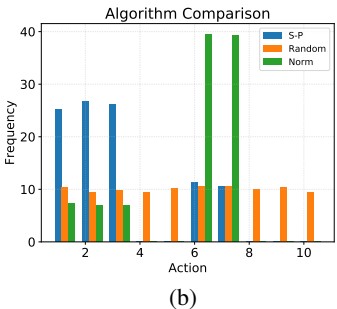

(a)                                (b)

Figure 2: Synthetic Setting. In the left figure, the policy value of the sampler-planner (S-P) is compared with a random algorithm (Random), a largest norm algorithm (Norm), and an algorithm that always chooses the same single action (Single). Supervised learning (SL) represents an approximate upper bound; however, we don't expect the algorithms to be able to reach the SL performance as they all receive bandit feedback compared to the full feedback available to SL (10,000 samples). In the right figure, the empirical distributions over the actions chosen by the algorithms are compared, with the exception of Single, which always chooses action 1.

We first conduct a synthetic validation experiment to verify our algorithms behave consistently with the intuition and theory previously described. We constructed a linear contextual bandit with $d = 20$ and $\mathcal{A} = \{1, \ldots, 10\}$. At each time, features were generated randomly through the following procedure. A category $I$ is sampled uniformly from the set $\{1, 2, 3\}$. For actions $a = 1, 2, 3$ and categories $i \in \{1, 2, 3\}$, we defined diagonal covariance matrices $\Sigma_{a,i} \in \mathbb{R}^{d \times d}$ such that $\Sigma_{a,i}$ has a single coordinate (determined by the pair $(a, i)$) along the diagonal equal to 1 while others are equal to $10^{-9}$. Given the category $I$, features are distributed as $\phi(s, a) \sim \mathcal{N}(0, \Sigma_{a,I})$. This ensures variance only in certain directions depending on the category sampled and action taken. Independent of the category, for actions $a = 6, 7$, features are distributed as $\phi(s, a) \sim \mathcal{N}(0, \mathrm{diag}(10^{-9}, \ldots, 10^{-9}, 5))$ independently. The last coordinate has large variance regardless of the other actions. Other actions yield features that are zero. The first $d - 1$ coordinates of $\theta$ were chosen randomly from $\{-1, 1\}^{d-1}$ and the $d$-th coordinate is zero.

We considered four algorithms: (1) a random algorithm that chooses actions uniformly from $\mathcal{A}$, (2) a largest norm algorithm that chooses the feature with the largest norm, (3) an algorithm that chooses action 1 and (4) our sampler-planner algorithm. The planner uses an independent context-only dataset of the same size as the one to be used for training. All algorithms are then run in the online phase with reward feedback. The algorithms are evaluated based on their resulting policy values during the online phase on a held-out test set. Policy value is defined as $\mathbb{E}_s \phi(s, \hat{\pi}(s))^\top \theta^\star$ where $\hat{\pi}$ is the learned policy for a given algorithm. All algorithms used $\lambda = 1$ and the planner used $\alpha = 1$ as these worked well. See appendix for more $\lambda$ settings.

Figure 2 shows the policy values plotted over the number of samples taken and also the action distributions of the algorithms. Each line/bar represents the mean/standard error of 20 trials. Most actions are designed to be uninformative in this setting and it is beneficial to select judiciously from actions $\{1, 2, 3, 6, 7\}$. A random algorithm should and does perform poorly. The largest norm algorithm also fails: while much is learned about the last coordinate of $\theta$ by typically taking actions $\{6, 7\}$, little is learned about the others which are necessary for making good decisions. Finally the single action algorithm fails to account for the categories.

## 7.2 Learning to rank dataset

We evaluate the planner-sampler performance on real-world data from the Yahoo! Learning to Rank challenge (Chapelle and Chang, 2011), which has been used previously (Foster et al., 2018).

The ranking dataset is structured as follows. Each datapoint (row) is a 700-dimensional feature vector associated with a ranking relevance score that takes on values in $\{0, 1, 2, 3, 4\}$. Each datapoint is also associated with a single context, termed a query. Multiple datapoints matched with the same query correspond to actions (documents) that the learner can choose when presented with that query. The dataset is already divided into training, validation, and testing data.

The objective is to choose actions, among those presented for a given query, that maximize the relevance scores. Unlike the previous synthetic dataset, the reward function here is misspecified, i.e., it does not exactly follow a linear model. To reduce the computational burden, we randomly subsampled coordinates of the original 700-dimensional features down to 300-dimensional features. We then normalized them to ensure the norms are at most 1. Similar to prior work, we also limit the number actions (documents) available at any given context to $K = 20$.

As before, we compare the planner-sampler pair to a random algorithm and a supervised learning oracle that observes all the context-action features and relevance scores, which is 200,000 training samples in total. To simulate streaming the data online to the algorithms, we sequentially iterate through the contexts and present the associated features to the learner as actions. During the offline phase, we run the planner on the validation set where none of the relevance scores are observed in order to generate the sampling policies. During the online phase, we run both the sampler and the random algorithm on the training data such that they observe the same contexts but may take different actions. The maximum number of distinct contexts we can iterate over is about 3000. The resulting policies are evaluated on the full test set after every 20 training ob-

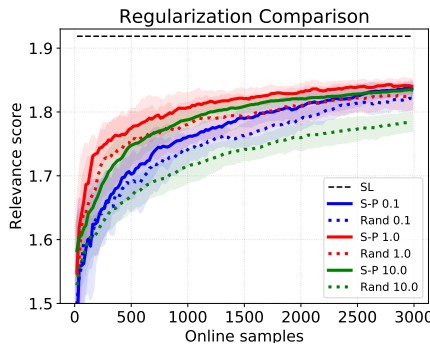

Figure 3: The sampler-planner (S-P) pair is compared with a random sampler (Rand) with varying levels of regularization $\lambda$. The supervised learning oracle (SL), which observes state-action pairs in the training set, is shown as baseline.

servations. This experiment was repeated with varying values of the regularization parameter $\lambda \in \{0.1, 1, 10\}$. The planner uses $\alpha = 1$; we did not find significant improvements with $\alpha < 1$. Additional values for $\lambda$ are reported in the supplementary material.

Each point/band on Figure 3 represents the mean/standard deviation of 10 independent trials. Randomness is due only to internal randomness in the algorithms and the randomized order in which the contexts are revealed. The sampler-planner outperforms the random algorithm in most regimes, including $\lambda = 1$ where the pinnacle performance of both algorithms is achieved. The gap between the algorithms is widened for larger choices of $\lambda$. For $\lambda = 0.1$ the curves appear to catch up as more data is gathered; it is possible that they would outperform a larger value of regularization if more samples were available. The optimal choice for $\lambda$ depends on the bias-variance tradeoff. Despite misspecification, our sampler-planner approach consistently matches or significantly exceeds the performance of the baseline approach, particularly when the number of samples is small.

## 8 Conclusion

Assuming access to a set of offline contexts, we presents an algorithm to find a single stochastic policy to acquire data strategically. The procedure is validated by a theoretical analysis and by experiments, even when the reward model is not linear. In the future it would be interesting to create an algorithm that works across all values of regularization without impacting the amount of offline context data required, and to develop methods to incorporate prior reward data to reduce the sample complexity of the online data needed to learn a good policy, perhaps through instance-dependent bounds that quantify the benefit of particular prior context-action-reward data in a specific domain.

Our work presents little direct negative potential societal impacts though whether contextualized decision policies are beneficial or harmful depends on the setting and the reward function chosen.

## Acknowledgments

This work is supported in part by NSF Grant #2112926 and GFRP (to JL), and a Toyota gift.

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
