# Contents

## A  Sample Complexity Results from Existing Literature

In this section we discuss how to adapt existing linear bandit results (mostly in the regret setting) to sample complexity results.

### A.1  Sample Complexity Lower Bounds

For the large action regime (i.e., $\min\{\ln\sum_s|\mathcal{A}_s|, d\} = d$), the lower bound $\approx d^2/\epsilon^2$ follows by using the same hypercube construction as Lattimore and Szepesvári (2020, Theorem 24.1).

For the small action regime (i.e., $\min\{\ln\sum_s|\mathcal{A}_s|, d\} = \ln\sum_s|\mathcal{A}_s|$). If $O(|A|) = d$, and we ignore the $\ln\sum_s|\mathcal{A}_s|$ term in sample complexity, the lower bound $\approx d/\epsilon^2$ follows by considering a bandit instance with $(d)$ orthogonal arms in $\mathbb{R}^d$. This is equivalent to a multi armed bandit problem, and the sample complexity lower bound then follows from Mannor and Tsitsiklis (2004, Theorem 1).

### A.2  Sample Complexity Upper Bounds

The upper bounds come from an instantiations of a general reduction in Jin et al. (2018, Section 3.1).

**Lemma 3** (Regret to sample complexity). *Running the algorithm in Abbasi-Yadkori et al. (2011) for $\Omega(\frac{d^2}{\epsilon^2})$ steps induces a $\epsilon$-optimal policy.*

*Proof.* First of all, Abbasi-Yadkori et al. (2011) achieve a regret after $N$ timesteps of at most $d\sqrt{N}$, meaning that $\sum_{t=1}^N (r(s_t, \pi^\star(s_t)) - r(s_t, \pi_t(s_t))) \leq O(d\sqrt{N})$.

Note that $\pi_t$ is independent of $s_t$, by the Azuma martingale concentration bound we get $\sum_{t=1}^N (\mathbb{E}_s r(s, \pi^\star(s)) - \mathbb{E}_s r(s, \pi_t(s))) \leq O(d\sqrt{N})$. Consider the average policy $\pi(s) = \pi_t(s)$ w.p. $1/N$. We get $\mathbb{E}_\pi \mathbb{E}_s[r(s, \pi(s))] \geq \mathbb{E}_s[r(s, \pi^\star(s))] - O(d/\sqrt{N})$. So when $N \geq \Omega(\frac{d^2}{\epsilon^2})$ the average policy played from timestep 1 to $N$ is at most $\epsilon$-suboptimal. $\square$

## B  Proof of Theorem 1

*Proof.* We start by invoking Proposition 2 *(Linear Regression with Union Bound)*. Define a short-hand for the RHS of proposition 2

$$\sqrt{\beta} \overset{def}{=} \min\{\alpha_1, \alpha_2\} + \sqrt{\lambda_{reg}}\|\theta^\star\|_2 \tag{13}$$

With a rescaling on $\delta$ we write

$$\mathbf{P}\left(\underbrace{\forall(s,a): \quad |\phi(s,a)^\top(\theta^\star - \widehat{\theta})| \lesssim \|\phi(s,a)\|_{(\Sigma_N')^{-1}}\ln(N)\sqrt{\beta}}_{\mathcal{E}_1}\right) \geq 1 - \frac{\delta}{4}. \tag{14}$$

Notice that under the event $\mathcal{E}_1$ above

$$\mathbb{E}_{s\sim\mu}\max_{a\in\mathcal{A}_s}|\phi(s,a)^\top(\theta^\star - \widehat{\theta})| \leq \ln(N)\sqrt{\beta}\,\mathbb{E}_{s\sim\mu}\max_{a\in\mathcal{A}_s}\|\phi(s,a)\|_{(\Sigma_N')^{-1}} = \ln(N)\sqrt{\beta}u_N'$$

Recall that $K$ is the number of policy switches by the sampler and Lemma 16 gives $K = \widetilde{O}(d)$. We set

$$\lambda_{reg} = \Omega(\ln(d/\delta)), \tag{15}$$

$$M = \Omega\left(\frac{KN}{\lambda_{reg}} \ln \frac{d^2 N}{\lambda_{reg}\delta}\right), \tag{16}$$

then Lemma 4 *(Relations between Offline and Online Uncertainty)* relates the expected online uncertainty $u'_N$ with the expected offline uncertainty $u_M$

$$\mathbf{P}\left(\underbrace{u'_N \lesssim u_M}_{\mathcal{E}_2}\right) \geq 1 - \frac{\delta}{2} \tag{17}$$

Finally, the expected offline uncertainty is bounded by in Lemma 5 *(Offline Expected Uncertainty)* chained with Lemma 7 *(Sum of Observed Uncertainties)*

$$\mathbf{P}\left(\underbrace{u_M \lesssim \frac{1}{M} \ln \frac{1}{\delta} + \sqrt{\frac{1}{\alpha M} d \ln\left(\frac{d\lambda_{reg} + M}{d}\right)}}_{\mathcal{E}_3}\right) \geq 1 - \frac{\delta}{4}. \tag{18}$$

A union bound over the events $\mathcal{E}_1, \mathcal{E}_2, \mathcal{E}_3$ and chaining the statements now produces with probability at least $1 - \delta$

$$\mathbb{E}_{s\sim\mu} \max_{a\in\mathcal{A}_s} |\phi(s,a)^\top(\theta^\star - \widehat{\theta})| \lesssim \frac{\ln(N)\sqrt{\beta}}{M} \ln \frac{1}{\delta} + \ln(N)\sqrt{\frac{\beta}{\alpha M} d \ln\left(\frac{d\lambda_{reg} + M}{d}\right)} \tag{19}$$

$$\lesssim \frac{\ln(N)\sqrt{\beta}}{M} \ln \frac{1}{\delta} + \ln(N)\sqrt{\frac{\beta}{N} d \ln\left(\frac{d\lambda_{reg} + M}{d}\right)} \tag{20}$$

after recalling $\alpha = N/M$. As a result, we set

$$N = \Omega\left(\frac{d\beta}{\epsilon^2} \ln^3\left(\lambda_{reg} + \frac{d^2\beta}{\lambda_{reg}\epsilon\delta}\right)\right)$$

and

$$M = \Omega\left(\frac{Kd\beta}{\lambda_{reg}\epsilon^2} \ln^4\left(\lambda_{reg} + \frac{d^2\beta}{\lambda_{reg}\epsilon\delta}\right)\right).$$

As a result, we satisfy the preconditions in Eq. (16) because

$$\frac{KN}{\lambda_{reg}} \ln \frac{d^2 N}{\lambda_{reg}\delta}$$

$$= \frac{K\frac{d\beta}{\epsilon^2} \ln^3\left(\lambda_{reg} + \frac{d^2\beta}{\lambda_{reg}\epsilon\delta}\right)}{\lambda_{reg}} \ln\left(\frac{d^2}{\lambda_{reg}\delta} \frac{d\beta}{\epsilon^2} \ln^3\left(\lambda_{reg} + \frac{d^2\beta}{\lambda_{reg}\epsilon\delta}\right)\right)$$

$$\lesssim \frac{Kd\beta}{\lambda_{reg}\epsilon^2} \ln^3\left(\lambda_{reg} + \frac{d^2\beta}{\lambda_{reg}\epsilon\delta}\right) 3\ln\left(\frac{d}{\lambda_{reg}\delta} \frac{d\beta}{\epsilon} \ln\left(\lambda_{reg} + \frac{d^2\beta}{\lambda_{reg}\epsilon\delta}\right)\right) \quad \text{(Note that } \frac{d\beta}{\lambda_{reg}\delta} \geq 1\text{)}$$

$$\lesssim \frac{Kd\beta}{\lambda_{reg}\epsilon^2} \ln^4\left(\lambda_{reg} + \frac{d^2\beta}{\lambda_{reg}\epsilon\delta}\right).$$

Note that $\epsilon \leq Kd\sqrt{\beta}/\lambda_{reg}$. Then

$$\frac{\ln(N)\sqrt{\beta}}{M} \ln \frac{1}{\delta} \lesssim \frac{\epsilon \ln(N)}{\ln^4\left(\lambda_{reg} + \frac{d^2\beta}{\lambda_{reg}\epsilon\delta}\right)} \ln \frac{1}{\delta} \tag{21}$$

By the facts $\ln(x\ln(y)) \leq \ln(xy) \leq 2\ln(\max(x,y))$ and $\ln(x^k) = k\ln(x)$, we have

$$\ln(N) = \ln\left(\frac{d\beta}{\epsilon^2} \ln^3\left(\lambda_{reg} + \frac{d^2\beta}{\lambda_{reg}\epsilon\delta}\right)\right) \leq 3\ln\left(\frac{d\beta}{\epsilon} \ln\left(\lambda_{reg} + \frac{d^2\beta}{\lambda_{reg}\epsilon\delta}\right)\right) \tag{22}$$

$$\leq 6 \ln \left( \max \left( \frac{d\beta}{\epsilon}, \lambda_{reg} + \frac{d^2\beta}{\lambda_{reg}\epsilon\delta} \right) \right) \leq 6 \ln \left( \lambda_{reg} + \frac{d^2\beta}{\lambda_{reg}\epsilon\delta} \right) ., \tag{23}$$

since $\lambda$ is at most $d$ so $\frac{d}{\lambda\delta} \geq 1$. Continuing Eq. (21), we get

$$\frac{\ln(N)\sqrt{\beta}}{M} \ln \frac{1}{\delta} \lesssim \frac{\epsilon \ln(N)}{\ln^4 \left( \lambda_{reg} + \frac{d^2\beta}{\lambda_{reg}\epsilon\delta} \right)} \ln \frac{1}{\delta} \lesssim \epsilon. \tag{24}$$

Note also that under this choice for $N$ the right most term of Eq. (20) can be upper bounded by

$$\ln(N)\sqrt{\frac{\beta}{N} d \ln \left( \frac{d\lambda_{reg} + M}{d} \right)} \lesssim \epsilon \ln(N) \sqrt{\ln^{-3} \left( \lambda_{reg} + \frac{d^2\beta}{\lambda_{reg}\epsilon\delta} \right) \ln \left( \frac{d\lambda_{reg} + M}{d} \right)}. \tag{25}$$

Recall that $K = \widetilde{O}(d) \leq O(d^2)$. The logarithmic term can be upper bounded by

$$\ln \left( \frac{d\lambda_{reg} + M}{d} \right) = \ln \left( \lambda_{reg} + \frac{K\beta}{\lambda_{reg}\epsilon^2} \ln^4 \left( \lambda_{reg} + \frac{d^2\beta}{\lambda_{reg}\epsilon\delta} \right) \right) \tag{26}$$

$$\lesssim \ln \left( \lambda_{reg} + \frac{d^2\beta}{\lambda_{reg}\epsilon^2} \ln^4 \left( \lambda_{reg} + \frac{d^2\beta}{\lambda_{reg}\epsilon\delta} \right) \right) \tag{27}$$

$$\leq 4 \ln \left( \lambda_{reg} + \frac{d^2\beta}{\lambda_{reg}\epsilon} \ln \left( \lambda_{reg} + \frac{d^2\beta}{\lambda_{reg}\epsilon\delta} \right) \right) \tag{28}$$

$$\leq 8 \ln \left( \lambda_{reg} + \frac{d^2\beta}{\lambda_{reg}\epsilon\delta} \right). \tag{29}$$

Now we continue with Eq. (25)

$$\epsilon \ln(N) \sqrt{\ln^{-3} \left( \lambda_{reg} + \frac{d^2\beta}{\lambda_{reg}\epsilon\delta} \right) \ln \left( \frac{d\lambda_{reg} + M}{d} \right)}$$

$$\lesssim \epsilon \ln(N) \sqrt{\ln^{-2} \left( \lambda_{reg} + \frac{d^2\beta}{\lambda_{reg}\epsilon\delta} \right)}$$

$$\lesssim \epsilon \ln \left( \lambda_{reg} + \frac{d^2\beta}{\lambda_{reg}\epsilon\delta} \right) \sqrt{\ln^{-2} \left( \lambda_{reg} + \frac{d^2\beta}{\lambda_{reg}\epsilon\delta} \right)} \lesssim \epsilon. \qquad \text{(By Eq. (23))}$$

Rescaling $\epsilon$ by a constant, we ensure

$$\mathbf{P} \left( \mathbb{E}_{s\sim\mu} \max_{a\in\mathcal{A}_s} |\phi(s,a)^\top (\theta^\star - \widehat{\theta})| \leq \epsilon \right) \geq 1 - \delta. \tag{30}$$

Now we prove Eq. (6). Let $\pi^\star(s) \stackrel{def}{=} \arg\max_{a\in\mathcal{A}_s} \phi(s,a)^\top \theta^\star$. Define $\Delta(s) \stackrel{def}{=} \max_{a\in\mathcal{A}_s} |\phi(s,a)^\top (\theta^\star - \widehat{\theta})|$ for shorthand. It follows immediately that

$$\phi(s,\widehat{\pi}(a))^\top \theta^\star \geq \phi(s,\widehat{\pi}(a))^\top \widehat{\theta} - \Delta(s), \tag{31}$$

$$\phi(s,\pi^\star(a))^\top \theta^\star \leq \phi(s,\pi^\star(a))^\top \widehat{\theta} + \Delta(s). \tag{32}$$

By the definition of $\widehat{\pi}$, we have

$$\phi(s,\widehat{\pi}(a))^\top \widehat{\theta} \geq \phi(s,\pi^\star(a))^\top \widehat{\theta}. \tag{33}$$

Combining Eqs. (31), (32), and (33) we get

$$\phi(s,\widehat{\pi}(a))^\top \theta^\star \geq \phi(s,\pi^\star(a))^\top \theta^\star - 2\Delta(s). \tag{34}$$

Consequently,

$$\mathbb{E}_{s\sim\mu} \left[ (\phi(s,\pi^\star(a)) - \phi(s,\pi(a)))^\top \theta^\star \right] \leq 2 \mathbb{E}_{s\sim\mu} [\Delta(s)]. \tag{35}$$

As a result, under the same event indicated by Eq. (30), we get

$$\mathbb{E}_{s\sim\mu} \left[ (\phi(s,\pi^\star(s)) - \phi(s,\widehat{\pi}(s)))^\top \theta^\star \right] \leq 2\epsilon, \tag{36}$$

which is exactly Eq. (6). $\qquad\square$

## C   Linear Regression

**Proposition 1** (Linear Regression Non-Adaptive Setting). *Consider drawing $n$ i.i.d. copies of $\phi_i$ from some fixed distribution, and define*

$$\widehat{\theta} = \left( \sum_{i=1}^{n} \phi_i \phi_i^\top + \lambda_{reg} I \right)^{-1} \sum_{i=1}^{n} \phi_i y_i$$

*where*

$$y_i = \phi_i^\top \theta^\star + \eta_i \tag{37}$$

*for some fixed $\theta^\star$ and $\eta_i$ is mean zero 1-subgaussian conditioned on $\phi_i$. Then for any fixed vector $x$*

$$\mathbf{P}\left( x^\top(\theta^\star - \widehat{\theta}) \leq \|x\|_{\Sigma^{-1}} \left( \sqrt{2 \ln \frac{2}{\delta}} + \sqrt{\lambda_{reg}} \|\theta^\star\|_2 \right) \right) \geq 1 - \delta.$$

*Proof.*

$$x^\top(\theta^\star - \widehat{\theta}) = x^\top \theta^\star - x^\top \left( \sum_{i=1}^{n} \phi_i \phi_i^\top + \lambda_{reg} I \right)^{-1} \left( \sum_{i=1}^{n} \phi_i \left( \phi_i^\top \theta^\star + \eta_n \right) + \lambda_{reg} \theta^\star - \lambda_{reg} \theta^\star \right) \tag{38}$$

$$= -x^\top \Sigma^{-1} \sum_{i=1}^{n} \phi_i \eta_i + \lambda_{reg} x^\top \Sigma^{-1} \theta^\star. \tag{39}$$

Using Cauchy-Schwartz we have

$$\lambda_{reg} x^\top \Sigma^{-1} \theta^\star \leq \lambda_{reg} \|x\|_{\Sigma^{-1}} \|\theta^\star\|_{\Sigma^{-1}}. \tag{40}$$

We have that $\lambda_{reg} \|\theta^\star\|_{\Sigma^{-1}} \leq \sqrt{\lambda_{reg}} \|\theta^\star\|_2$. Since the feature vectors $\phi_i$'s are sampled from a fixed distribution, conditioned on the sampled state-actions $\phi_i$, both the covariance matrix $\Sigma$ is fixed and the noise $\eta_i$ is independent and 1 sub-Gaussian. Define $v_i = x^T \Sigma^{-1} \phi_i \eta_i$. Then conditioned on the sampled state-actions $\phi_i$, the $v$s are independent random $(x^T \Sigma^{-1} \phi_i)^2$ sub-Gaussian random variables and we can apply Hoeffding's inequality, conditioned on the observed state-action features $\phi_i$:

$$\mathbf{P}\left( |\sum_{i=1}^{n} x^\top \Sigma^{-1} \phi_i \eta_i| \leq \sqrt{2 \sum_{i=1}^{n} (x^\top \Sigma^{-1} \phi_i)^2 \ln \frac{2}{\delta}} = \|x\|_{\Sigma^{-1}} \sqrt{2 \ln \frac{2}{\delta}} \right) \geq 1 - \delta. \tag{41}$$

Combining with the regularization part, we conclude. $\qquad\square$

**Proposition 2** (Linear Regression with Union Bound). *In the same setting as proposition 1 we have that*

$$\mathbf{P}\left( \forall (s,a) : \phi(s,a)^\top(\theta^\star - \widehat{\theta}) \leq \|\phi(s,a)\|_{\Sigma^{-1}} \left( \min\{\alpha_1, \alpha_2\} + \sqrt{\lambda_{reg}} \|\theta^\star\|_2 \right) \right) \geq 1 - \delta.$$

*where*

$$\alpha_1 \overset{def}{=} \sqrt{2 \ln 2 |\sum_s \mathcal{A}_s| + \ln \frac{1}{\delta}} \tag{42}$$

$$\alpha_2 \overset{def}{=} 2\sqrt{2d \ln 6 + \ln \frac{1}{\delta}} \tag{43}$$

*Proof.* The proof essentially follows from taking the sharper of two results obtained as follows:

1. Invoke proposition 1 with a union bound over the state-action space (this gives the bound with the $\alpha_1$ term)

2. This follows from a discretization argument; see e.g., (Lattimore and Szepesvári, 2020), in particular, their equation 20.3.

$\qquad\square$

# D    Bounding the Uncertainty

We use the notation $u_m = \mathbb{E}_m \, U_m$ and $u'_n = \mathbb{E}'_n \, U'_n$ for short.

**Lemma 4** (Relations between Offline and Online Uncertainty). *If $\lambda_{reg} = \Omega(\ln \frac{d}{\delta})$ and $M = \Omega\left(\frac{KN}{\lambda_{reg}} \ln \frac{dNK}{\lambda_{reg}\delta}\right)$, upon termination of algorithms 1 and 2 it holds that*

$$\mathbf{P}\left(u'_N \lesssim u_M\right) \geq 1 - \frac{\delta}{2}$$

*where the probability is over the offline context dataset $\mathcal{C} = \{s_1, \ldots, s_M\}$ and the online context dataset $\mathcal{C}' = \{s'_1, \ldots, s'_N\}$.*

*Proof.* Let $d_1, \ldots, d_M$ be the conditional distributions of the feature vectors sampled at timesteps $1, \ldots, n$ in algorithm 1 after the algorithm has terminated. Conditioned on $\mathcal{F}_M = \sigma(s_1, \ldots, s_M)$, the $d_i$'s are non-random; let

$$\overline{\Sigma} = \alpha \sum_{i=1}^{M} \mathbb{E}_{\phi \sim d_i} \, \phi\phi^\top + \lambda_{reg} I \tag{44}$$

be the conditional expectation of the cumulative covariance matrix and let

$$\Sigma'_n = \sum_{i=1}^{n} (\phi'_i)(\phi'_i)^\top + \lambda_{reg} I \tag{45}$$

be the cumulative covariance matrix experienced in algorithm 2 where $\phi'_i \stackrel{def}{=} \phi(s'_n, a'_n)$ is the sampled feature during the execution of algorithm 1. If $\lambda_{reg} = \Omega(\ln \frac{d}{\delta})$ and $M = \Omega\left(\frac{KN}{\lambda_{reg}} \ln \frac{dNK}{\lambda_{reg}\delta}\right)$ we obtain from Lemma 14 *(Matrix Upper Bound Offline Phase)* and Lemma 15 *(Matrix Upper Bound Online Phase)*

$$\mathbf{P}\left(\forall x, \|x\|_2 \leq 1 : \|x\|_{\overline{\Sigma}^{-1}} \leq 3\,\|x\|_{\Sigma_M^{-1}}\right) \geq 1 - \frac{\delta}{4} \tag{46}$$

$$\mathbf{P}\left(\forall x, \|x\|_2 \leq 1 : \|x\|_{(\Sigma'_N)^{-1}} \leq 9\,\|x\|_{\overline{\Sigma}^{-1}}\right) \geq 1 - \frac{\delta}{4}. \tag{47}$$

Define the policy maximizing the online uncertainty

$$\pi'_n(s) = \underset{a \in \mathcal{A}_s}{\arg\max} \, \|\phi(s, a)\|_{(\Sigma'_n)^{-1}}. \tag{48}$$

Under the two above events we can write

$$u'_N \stackrel{def}{=} \mathbb{E}_{s\sim\mu} \max_{a \in \mathcal{A}_s} \|\phi(s, a)\|_{(\Sigma'_N)^{-1}} \tag{49}$$

$$= \mathbb{E}_{s\sim\mu} \|\phi(s, \pi'_N(s))\|_{(\Sigma'_N)^{-1}} \tag{50}$$

$$\leq 9\,\mathbb{E}_{s\sim\mu} \|\phi(s, \pi'_N(s))\|_{\overline{\Sigma}^{-1}} \tag{51}$$

$$\leq 27\,\mathbb{E}_{s\sim\mu} \|\phi(s, \pi'_N(s))\|_{\Sigma_M^{-1}} \tag{52}$$

$$\leq 27\,\mathbb{E}_{s\sim\mu} \max_a \|\phi(s, a)\|_{\Sigma_M^{-1}} \tag{53}$$

$$\leq 27\,\mathbb{E}_{s\sim\mu} \max_a \|\phi(s, a)\|_{\Sigma_M^{-1}} \tag{54}$$

$$= 27 u_M. \tag{55}$$

$\square$

**Lemma 5** (Offline Expected Uncertainty). *We have that*

$$\mathbf{P}\left(u_M \leq \frac{1}{M} \sum_{m=1}^{M} u_m \lesssim \frac{1}{M}\left[\ln \frac{1}{\delta} + \sum_{m=1}^{M} U_m\right] \stackrel{def}{=} \frac{\mathcal{R}(M, \frac{\delta}{4})}{M}\right) \geq 1 - \frac{\delta}{4}.$$

*Proof.* Consider the event that the sum of the predictable means $\sum_{m=1}^{M} u_m = \sum_{m=1}^{M} \mathbb{E}[U_m \mid \mathcal{F}_m]$ does not deviate significantly from $\sum_{m=1}^{M} U_m$:

$$\mathcal{E}(\delta) \stackrel{def}{=} \left\{ \sum_{n=1}^{M} u_m \leq \frac{1}{4} \left( c_1(\delta) + \sqrt{c_1^2(\delta) + 4 \left( \sum_{m=1}^{M} U_m + c_2(\delta) \right)} \right)^2 \stackrel{def}{=} \mathcal{R}hs(M, \delta) \right\}. \quad (56)$$

Using Theorem 3 *(Reverse Bernstein for Martingales)* (which also defines $c_1, c_2$) we obtain

$$\mathbf{P}\left(\mathcal{E}(\delta/4)\right) \geq 1 - \frac{\delta}{4}. \quad (57)$$

From Lemma 6 *(Decreasing Uncertainty)* we know that the sequence $\{u_n\}_{n=1}^{N+1}$ is surely decreasing, which means that the last element must be less than the average:

$$u_M \leq \frac{1}{M} \sum_{n=1}^{M} u_m = \frac{\mathcal{R}hs(M, \delta/4)}{M} \quad (58)$$

where in particular the equality holds under the event $\mathcal{E}(\delta)$. Finally using Cauchy-Schwartz we conclude that under the same event

$$\sum_{n=1}^{M} u_m \lesssim c_1^2(\delta) + c_2(\delta) + \sum_{m=1}^{M} U_m \stackrel{def}{=} \mathcal{R}(M, \delta). \quad (59)$$

$\square$

**Lemma 6** (Decreasing Uncertainty). *For every $n$ it holds that*

$$u_{n+1} \leq u_n.$$

*Proof.* By linear algebra, we must have

$$\Sigma_{n+1} \succeq \Sigma_n \quad \longrightarrow \quad \Sigma_{n+1}^{-1} \preceq \Sigma_n^{-1}$$

Using the definitions we have:

$$u_{n+1} = \mathbb{E}_{s \sim \mu} \max_{a \sim \mathcal{A}_s} \|\phi(s, a)\|_{\Sigma_{n+1}^{-1}} \quad (60)$$

$$\leq \mathbb{E}_{s \sim \mu} \max_{a \sim \mathcal{A}_s} \|\phi(s, a)\|_{\Sigma_n^{-1}} \quad (61)$$

$$= u_n. \quad (62)$$

$\square$

**Lemma 7** (Sum of Observed Uncertainties). *If $\lambda_{reg} \geq 1$ and and $\alpha \leq 1$ then*

$$D \stackrel{def}{=} \sum_{n=1}^{M} U_m \lesssim \sqrt{\frac{M}{\alpha} d \ln \left( \frac{d\lambda_{reg} + M}{d} \right)}. \quad (63)$$

*Proof.* Let $\{\phi_m\}$ be the feature vectors experienced during the offline phase; we have

$$\sum_{m=1}^{M} U_m = \sum_{m=1}^{M} \|\phi_m\|_{\Sigma_m^{-1}} \quad (64)$$

$$= \sum_{m=1}^{M} \|\phi_m\|_{\left(\alpha \sum_{j=1}^{m} \phi_j \phi_j + \lambda_{reg} I\right)^{-1}} \quad (65)$$

$$\leq \sqrt{M \sum_{m=1}^{M} \|\phi_m\|^2_{\left(\alpha \sum_{j=1}^{m} \phi_j \phi_j + \lambda_{reg} I\right)^{-1}}} \quad (66)$$

$$= \sqrt{\frac{M}{\alpha} \sum_{m=1}^{M} \|\sqrt{\alpha}\phi_m\|^2_{(\sum_{j=1}^{m} \sqrt{\alpha}\phi_j \sqrt{\alpha}\phi_j + \lambda_{reg}I)^{-1}}} \tag{67}$$

$$\leq \sqrt{\frac{M}{\alpha} \times 3 \times \underbrace{\ln \frac{\det\left(\sum_{j=1}^{M} \sqrt{\alpha}\phi_j \sqrt{\alpha}\phi_j^\top + \lambda_{reg}I\right)}{\det(\lambda_{reg}I)}}_{\stackrel{def}{=}\mathcal{I}}} \tag{68}$$

where the first inequality follows from Cauchy-Schwartz. The second inequality follows from Lemma 8 *(Elliptical Potential Argument Lemma with Doubling)*, where the precondition for Lemma 8 *(Elliptical Potential Argument Lemma with Doubling)* is satisfied by Lemma 9 *(Maximum Determinant Ratio)* since $\alpha \leq 1$. Finally, to bound the information gain $\mathcal{I}$, note $||\sqrt{\alpha}\phi_i||_2 \leq 1$ since $\alpha \leq 1$. Then (Abbasi-Yadkori et al., 2011)'s Lemma 11

$$\ln \det \left(\sum_{j=1}^{M} \sqrt{\alpha}\phi_j \sqrt{\alpha}\phi_j^T + \lambda_{reg}I\right) - \ln \det(\lambda_{reg}I))$$

$$\leq d\ln((\text{Tr}(\lambda_{reg}I) + M)/d) - \ln\det(\lambda_{reg}I)$$

Since

$$\ln \det(\lambda_{reg}I) = d\ln(\lambda_{reg}) \geq 1. \tag{69}$$

Then

$$\ln \det \left(\sum_{j=1}^{M} \sqrt{\alpha}\phi_j \sqrt{\alpha}\phi_j^\top + \lambda_{reg}I\right) \leq d\ln\left(\frac{d\lambda_{reg} + M}{d}\right) \tag{70}$$

$\square$

**Lemma 8** (Elliptical Potential Argument Lemma with Doubling)**.** *(see (Zanette et al., 2021, Lemma 36)) Let $x_1, \cdots, x_M$ be a sequence of vectors such that $\|x_i\|_2 \leq 1$. Define $\Sigma_m = \lambda_{reg}I + \sum_{i=1}^{m-1} x_i x_i^\top$. Suppose $\underline{m} \leq m$ satisfies $\det(\Sigma_m) \leq 4\det(\Sigma_{\underline{m}})$. Then we have*

$$\sum_{m=1}^{M} \|x_i\|_{\Sigma_{\underline{m}}^{-1}} \leq 3\ln \frac{\det(\Sigma_{M+1})}{\det(\lambda_{reg}I)}. \tag{71}$$

**Lemma 9** (Maximum Determinant Ratio)**.** *(see (Zanette et al., 2021, Lemma 34)) Let $x_1, \cdots, x_M$ be a sequence of vectors such that $\|x_i\|_2 \leq 1$. and assume $\lambda \geq 1$. Define $\Sigma_m = \lambda_{reg}I + \sum_{i=1}^{m-1} x_i x_i^\top$. Then for $\underline{m} \leq m$ we have $\det(\Sigma_m) \leq 4\det(\Sigma_{\underline{m}})$.*

# E    Matrix Concentration Inequalities

In this section we present matrix concentration inequalities used in our proof.

## E.1    Known Matrix Concentration Inequalities

The following result about eigenvalues lower and upper bounds is well known.

**Lemma 10** (Theorem 1.1 of Tropp (2012))**.** *Let $X_1, \cdots, X_n$ be a sequence of independent, positive semi-definite, self-adjoint matrices with dimension $d$. Suppose $\lambda_{\max}(X_k) \leq R$ almost surely. Define $\mu_{\min} = \lambda_{\min}(\sum_k \mathbb{E} X_k)$ and $\mu_{\max} = \lambda_{\max}(\sum_k \mathbb{E} X_k)$. Then*

$$\Pr\left(\lambda_{\min}\left(\sum_{k=1}^{n} X_k\right) \leq (1-\delta)\mu_{\min}\right) \leq d\left(\frac{e^{-\delta}}{(1-\delta)^{1-\delta}}\right)^{\mu_{\min}/R} \quad \text{for } \delta \in [0,1), \tag{72}$$

$$\Pr\left(\lambda_{\max}\left(\sum_{k=1}^{n} X_k\right) \geq (1+\delta)\mu_{\max}\right) \leq d\left(\frac{e^{\delta}}{(1+\delta)^{1+\delta}}\right)^{\mu_{\max}/R} \quad \text{for } \delta \geq 0. \tag{73}$$

We loosen the above inequalities to make them more amenable to a direct use.

**Corollary 1.** *In the setting of Lemma 10, with $\delta \in [0, 1]$*

$$\Pr\left(\lambda_{\min}\left(\sum_{k=1}^n X_k\right) \leq (1-\delta)\mu_{\min}\right) \leq d\left(1 - \frac{\delta^2}{2}\right)^{\mu_{\min}/R}, \tag{74}$$

$$\Pr\left(\lambda_{\max}\left(\sum_{k=1}^n X_k\right) \geq (1+\delta)\mu_{\max}\right) \leq d\left(1 - \frac{\delta^2}{4}\right)^{\mu_{\max}/R}. \tag{75}$$

*In addition, for any $\mu \geq \mu_{\max}$,*

$$\Pr\left(\lambda_{\max}\left(\sum_{k=1}^n X_k\right) \geq 2\mu\right) \leq d\exp\left(-\mu/(4R)\right). \tag{76}$$

*Proof.* The first two inequalities follows from the fact that $\forall \delta \in [0, 1]$

$$\frac{e^\delta}{(1+\delta)^{1+\delta}} \leq 1 - \frac{\delta^2}{4} \tag{77}$$

$$\frac{e^{-\delta}}{(1-\delta)^{1-\delta}} \leq 1 - \frac{\delta^2}{2}. \tag{78}$$

Now we prove Eq. (76). Let $r = \mu/\mu_{\max}$. Since $r \geq 1$ we have

$$\Pr\left(\lambda_{\max}\left(\sum_{k=1}^n X_k\right) \geq (1+\delta)\mu\right) = \Pr\left(\lambda_{\max}\left(\sum_{k=1}^n X_k\right) \geq (r + r\delta)\mu_{\max}\right) \tag{79}$$

$$\leq \Pr\left(\lambda_{\max}\left(\sum_{k=1}^n X_k\right) \geq (1+r\delta)\mu_{\max}\right) \leq d\left(\frac{e^{r\delta}}{(1+r\delta)^{1+r\delta}}\right)^{\mu_{\max}/R}, \tag{80}$$

where the last inequality is due to Lemma 10.

By basic algebra we have $\frac{e^x}{(1+x)^{1+x}} \leq e^{-x/4}$ for all $x \geq 1$. As a result, let $\delta = 1$ we have

$$d\left(\frac{e^r}{(1+r)^{1+r}}\right)^{\mu_{\max}/R} \leq d\exp\left(-\frac{r\mu_{\max}}{4R}\right) = d\exp\left(-\frac{\mu}{4R}\right). \tag{81}$$

Therefore,

$$\Pr\left(\lambda_{\max}\left(\sum_{k=1}^n X_k\right) \geq 2\mu\right) \leq d\exp\left(-\frac{\mu}{4R}\right). \tag{82}$$

which completes the proof. $\qquad\square$

### E.2 Matrix Concentration Inequalities in All Directions

In the following development we need to 'sandwich' the cumulative matrix around its expectation; as a step towards this, we first derive concentration inequalities as a function of the minimum eigenvalue.

For the following lemma, see also Lemma 20 (Ruan et al., 2020).

**Lemma 11** (Matrix Upper and Lower Bound with Minimum Eigenvalue). *Let $X_1, \cdots, X_n \sim \mathcal{D}$ be i.i.d. samples from $\mathcal{D}$ where $X_k \in \mathbb{R}^{d \times d}$. Suppose $X_k$ is positive semi-definite for all $k \in [n]$ and $\lambda_{\max}(X_k) \leq 1$ almost surely. Let $\lambda = \lambda_{\min}(\mathbb{E}_{X \sim \mathcal{D}} X) > 0$. Then for $\delta \in [0, 1]$ we have*

$$\Pr\left(\frac{1}{n}\sum_{k=1}^n X_k \preccurlyeq (1+\delta)\mathbb{E}[X]\right) \geq 1 - d\left(1 - \frac{\delta^2}{4}\right)^{n\lambda} \tag{83}$$

$$\Pr\left(\frac{1}{n}\sum_{k=1}^n X_k \succcurlyeq (1-\delta)\mathbb{E}[X]\right) \geq 1 - d\left(1 - \frac{\delta^2}{2}\right)^{n\lambda}. \tag{84}$$

*Proof.* We prove the first inequality first. Let $\Sigma = \mathbb{E}[X]$ and define $Y = \Sigma^{-1/2} X \Sigma^{-1/2}$. Then using linearity of expectation we have

$$\mathbb{E}[Y_k] = \mathbb{E}[\Sigma^{-1/2} X_k \Sigma^{-1/2}] = \Sigma^{-1/2} \left( \mathbb{E}\, X_k \right) \Sigma^{-1/2} = \Sigma^{-1/2} \Sigma \Sigma^{-1/2} = I. \qquad (85)$$

In addition,

$$\|Y_k\|_{\mathrm{op}} \leq \left\| \Sigma^{-1/2} \right\|_{\mathrm{op}} \|X_k\|_{\mathrm{op}} \left\| \Sigma^{-1/2} \right\|_{\mathrm{op}} \leq 1/\lambda$$

almost surely. As a result, $\lambda_{\max}(Y_k^\top) \leq 1/\lambda$. Now corollary 1 gives (with $\mu_{max} = \mu_{min} = n$)

$$\Pr\left( \lambda_{\max}\left( \sum_{k=1}^n Y_k \right) \leq (1-\delta)n \right) \leq d \left( 1 - \frac{\delta^2}{2} \right)^{n\lambda}, \qquad (86)$$

$$\Pr\left( \lambda_{\max}\left( \sum_{k=1}^n Y_k \right) \geq (1+\delta)n \right) \leq d \left( 1 - \frac{\delta^2}{4} \right)^{n\lambda}. \qquad (87)$$

Now, to derive the result for $\delta \in [0,1]$ consider

$$\Pr\left( \frac{1}{n} \sum_{k=1}^n X_k \preccurlyeq (1+\delta)\, \mathbb{E}[X] \right) = \Pr\left( \frac{1}{n} \sum_{k=1}^n Y_k \preccurlyeq (1+\delta)I \right) \qquad \text{(By Eq.(85))}$$

$$= \Pr\left( \lambda_{\max}\left( \sum_{k=1}^n Y_k \right) \leq (1+\delta)n \right)$$

$$= 1 - \Pr\left( \lambda_{\max}\left( \sum_{k=1}^n Y_k \right) > (1+\delta)n \right)$$

$$\geq 1 - d \left( 1 - \frac{\delta^2}{4} \right)^{n\lambda}. \qquad \text{(By Eq. (86))}$$

Finally, to derive the other statement we proceed similarly. We have

$$\Pr\left( \frac{1}{n} \sum_{k=1}^n X_k \succcurlyeq (1-\delta)\, \mathbb{E}[X] \right) = \Pr\left( \frac{1}{n} \sum_{k=1}^n Y_k \succcurlyeq (1-\delta)I \right) \qquad \text{(By Eq.(85))}$$

$$= \Pr\left( \lambda_{\max}\left( \sum_{k=1}^n Y_k \right) \geq (1-\delta)n \right)$$

$$= 1 - \Pr\left( \lambda_{\max}\left( \sum_{k=1}^n Y_k \right) < (1-\delta)n \right)$$

$$\geq 1 - d \left( 1 - \frac{\delta^2}{2} \right)^{n\lambda}. \qquad \text{(By Eq. (87))}$$

$\square$

Using the lemma just derived, we can derive a matrix upper bound (in all directions) that does not depend on the minimum eigenvalue.

**Lemma 12** (Matrix Upper Bound). *Let $X_1, \cdots, X_n$ be i.i.d. samples from $\mathcal{D}$ where $X_k \in \mathbb{R}^{d \times d}$. Suppose $X_k$ is positive semi-definite and $\|X_k\|_{\mathrm{op}} \leq 1$ for all $k \in [n]$ almost surely. For any fixed $\lambda > 0$,*

$$\Pr\left( \frac{1}{n} \sum_{k=1}^n X_k \preccurlyeq 10\lambda I + 3\, \mathbb{E}_{X \sim \mathcal{D}}[X] \right) \geq 1 - 2d \exp\left( -\frac{n\lambda}{4} \right). \qquad (88)$$

*Proof.* Let $\Sigma = \mathbb{E}_{x \sim \mathcal{D}}[X]$. Consider the spectrum decomposition of $\Sigma$, denoted by $\Sigma = \sum_{k=1}^d \lambda_k v_k v_k^\top$. Without loss of generality, we assume $\lambda_1 \geq \lambda_2 \geq \cdots \geq \lambda_d$. Let $R = \sup\{k :$

$\lambda_k \geq \lambda\}$. Define $P_+ = \sum_{k=1}^R v_k v_k^\top$, $P_- = \sum_{k=R+1}^d v_k v_k^\top$ and $Q = \sum_{k=1}^R v_k e_k^\top \in \mathbb{R}^{d \times R}$ where $e_k \in \mathbb{R}^R$ is the $k$-th basis for $R$-dimensional space.

Since $\{v_k\}$ is a set of orthogonal basis, we have $P_+ + P_- = I$. By algebraic manipulation we get

$$\frac{1}{n} \sum_{k=1}^n X_k = (P_+ + P_-)^\top \left( \frac{1}{n} \sum_{k=1}^n X_k \right) (P_+ + P_-) \tag{89}$$

$$= \frac{5}{4n} \sum_{k=1}^n P_+^\top X_k P_+ - \frac{1}{n} \sum_{k=1}^n \left( \frac{1}{2} P_+ - 2 P_- \right)^\top X_k \left( \frac{1}{2} P_+ - 2 P_- \right) + \frac{5}{n} \sum_{k=1}^n P_-^\top X_k P_-. \tag{90}$$

Note that for any $u \in \mathbb{R}^R$ we have $u^\top Q^\top \Sigma Q u = \sum_{k=1}^R \lambda_k \langle u, e_k \rangle^2 \geq \lambda \|u\|_2^2$. As a result, $\lambda_{\min}(Q^\top \Sigma Q) = \lambda_{\min}(\mathbb{E}_{x \sim \mathcal{D}} Q^\top X Q) \geq \lambda$. Consequently, applying Lemma 11 we get

$$\Pr \left( \frac{1}{n} \sum_{k=1}^n P_+^\top X_k P_+ \preccurlyeq 2 \mathbb{E}_{X \sim \mathcal{D}}[P_+^\top X P_+] \right)$$

$$= \Pr \left( \frac{1}{n} \sum_{k=1}^n Q Q^\top X_k Q Q^\top \preccurlyeq 2 \mathbb{E}_{X \sim \mathcal{D}}[Q Q^\top X Q Q^\top] \right) \qquad \text{(By the definition of } Q.\text{)}$$

$$\geq \Pr \left( \frac{1}{n} \sum_{k=1}^n Q^\top X_k Q \preccurlyeq 2 \mathbb{E}_{X \sim \mathcal{D}}[Q^\top X Q] \right)$$

$$\geq 1 - d \left( 1 - \frac{1}{4} \right)^{n\lambda} \geq 1 - d \exp \left( -\frac{n\lambda}{4} \right).$$

[Note we cannot directly apply Lemma 11 to the top of the above sequence because that lemma requires a minimum eigenvalue greater than 0 and $P_+^\top X_k P_+$ is not full rank.]

Next, by the linearity of expectation we get $\mathbb{E}_{X \sim \mathcal{D}}[P_+ X P_+^\top] = P_+(\mathbb{E}_{X \sim \mathcal{D}}[X])P_+^\top = P_+ \Sigma P_+^\top$. Recall that the spectrum decomposition gives $\Sigma = \sum_{k=1}^d \lambda_k v_k v_k^\top$, where $\langle v_k, v_j \rangle = \mathbb{I}[j = k]$. As a result,

$$P_+^\top \Sigma P_+ = \left( \sum_{k=1}^R v_k v_k^\top \right) \left( \sum_{k=1}^d \lambda_k v_k v_k^\top \right) \left( \sum_{k=1}^R v_k v_k^\top \right) \tag{91}$$

$$= \left( \sum_{k=1}^R \lambda_k v_k v_k^\top \right) \preccurlyeq \left( \sum_{k=1}^d \lambda_k v_k v_k^\top \right) = \Sigma. \tag{92}$$

Consequently we get $\mathbb{E}_{X \sim \mathcal{D}}[P_+ X^\top P_+] = P_+^\top \Sigma P_+ \preccurlyeq \Sigma = \mathbb{E}_{X \sim \mathcal{D}}[X]$. Therefore we have

$$\Pr \left( \frac{1}{n} \sum_{k=1}^n P_+^\top X_k P_+ \preccurlyeq 2 \mathbb{E}_{X \sim \mathcal{D}}[X] \right) \geq 1 - d \exp \left( -\frac{n\lambda}{4} \right). \tag{93}$$

On the other hand, we upper bound the third term in Eq. (90) by Lemma 10. Let $Y_k = P_-^\top X_k P_-$. Then we have $\|Y_k\|_{\mathrm{op}} \leq \|X_k\|_{\mathrm{op}} \leq 1$. In addition,

$$\mathbb{E}[Y_k] = P_- \Sigma P_-^\top. \tag{94}$$

We claim that $\lambda_{\max}(\mathbb{E}[Y_k]) \leq \lambda$. Indeed, for any $v \in S^{d-1}$ we have

$$v^\top \mathbb{E}[Y_k] v = v^\top P_- \Sigma P_-^\top v. \tag{95}$$

Recall that $P_- = \sum_{k=1}^d \mathbb{I}[\lambda_k < \lambda] v_k v_k^\top$ where $\{\lambda_k\}, \{v_k\}$ are the eigen-vectors and eigen-values of $\Sigma$. For any $v \in S^{d-1}$, $P_-^\top v$ lies in the linear space spanned by $\{v_i : \lambda_i < \lambda\}$. As a result, $\|\Sigma P_-^\top v\|_2 \leq \lambda \|v\|_2$. Consequently,

$$v^\top \mathbb{E}[Y_k] v \leq \|P_-^\top v\|_2 \|\Sigma P_-^\top v\|_2 \leq \lambda \|v\|_2^2. \tag{96}$$

for all $v \in S^{d-1}$. Therefore we prove $\lambda_{\max}(\mathbb{E}[Y_k]) \leq \lambda$.

Now apply corollary 1 on $Y_k$ and we get

$$\Pr\left(\frac{1}{n}\sum_{k=1}^{n} Y_k \preccurlyeq 2\lambda I\right) \geq 1 - d\exp\left(-\frac{\lambda n}{4}\right). \tag{97}$$

Under the high probability events described by Eq. (93) and Eq. (97), with probability at least $1 - 2d\exp(-\lambda n/4)$ we get

$$\frac{1}{n}\sum_{k=1}^{n} X_k \tag{98}$$

$$=\frac{5}{4n}\sum_{k=1}^{n} P_+^\top X_k P_+ - \frac{1}{n}\sum_{k=1}^{n}\left(\frac{1}{2}P_+ - 2P_-\right)^\top X_k \left(\frac{1}{2}P_+ - 2P_-\right) + \frac{5}{n}\sum_{k=1}^{n} P_-^\top X_k P_- \tag{99}$$

$$\preccurlyeq \frac{5}{4n}\sum_{k=1}^{n} P_+^\top X_k P_+ + \frac{5}{n}\sum_{k=1}^{n} P_-^\top X_k P_- \tag{100}$$

$$\preccurlyeq \frac{5}{2}\mathbb{E}_{X\sim\mathcal{D}}[X] + 10\lambda I. \tag{101}$$

$\square$

Likewise, we can easily obtain the following matrix lower bound without any dependence on the minimum eigenvalue (see also Lemma 21 from (Ruan et al., 2020)).

**Lemma 13** (Matrix Lower Bound). *Let $X_1, \cdots, X_n$ be i.i.d. samples from $\mathcal{D}$ where $X_k \in \mathbb{R}^{d\times d}$. Suppose $X_k$ is positive semi-definite and rank one and $\|X_k\|_{\mathrm{op}} \leq 1$ for all $k \in [n]$ almost surely. For any fixed $\lambda \geq 0$*

$$\Pr\left(3\lambda I + \frac{1}{n}\sum_{k=1}^{n} X_k \succcurlyeq \frac{1}{8}\mathbb{E}_{X\sim\mathcal{D}}[X]\right) \geq 1 - 2d\exp\left(-\frac{n\lambda}{8}\right). \tag{102}$$

**Corollary 2.** *Let $X_1, \cdots, X_n$ be i.i.d. samples from $\mathcal{D}$ where $X_k \in \mathbb{R}^{d\times d}$. Suppose $X_k$ is positive semi-definite, and $\|X_k\|_{\mathrm{op}} \leq 1$ for all $k \in [n]$ almost surely. For any fixed $t > 0$ we have*

$$\Pr\left(\forall m \leq n, \sum_{k=1}^{m} X_k \preccurlyeq 10tI + 3m\,\mathbb{E}_{X\sim\mathcal{D}}[X]\right) \geq 1 - 2nd\exp\left(-t/4\right). \tag{103}$$

$$\tag{104}$$

*Proof.* For any fixed $m \leq n$, applying lemma 12 with $\lambda = t/m$ we get

$$\Pr\left(\sum_{k=1}^{m} X_k \preccurlyeq 10tI + 3m\,\mathbb{E}_{X\sim\mathcal{D}}[X]\right) = \Pr\left(\frac{1}{m}\sum_{k=1}^{m} X_k \preccurlyeq 10\frac{t}{m}I + 3\,\mathbb{E}_{X\sim\mathcal{D}}[X]\right) \tag{105}$$

$$\leq 1 - 2d\exp\left(-t/4\right). \tag{106}$$

By union bound over $m \in [n]$ we prove Eq. (103). $\square$

### E.3 Relation Between Offline and Online Covariance Matrices

**Notation**: Let $n_k$ be the expected number of samples in the online phase allocated to policy $\pi^k$.

**Lemma 14** (Matrix Upper Bound Offline Phase). *algorithm 1 produces a comulative covariance matrix $\Sigma_M$ that satisfies*

$$\mathbf{P}\left(\Sigma_M \preccurlyeq 3\overline{\Sigma}\right) \geq 1 - \frac{\delta}{4} \tag{107}$$

*as long as*

$$M \geq \frac{160KN}{\lambda_{reg}}\ln\frac{320dNK}{\lambda_{reg}\delta}. \tag{108}$$

*Proof.* From Lemma 16 *(Number of Switches)* we know that at most $K$ distinct policies are produced during the execution of algorithm 1 where $K$ is defined in that lemma.

Let $\phi_i^{(k)}$ be the $i$ sampled feature during phase $k$. Let $m_k$ be the values that $m$ takes on during phase $k$. Note that although $m_k$ is a random variable, we have $m_k \leq M$ almost surely. As a result, applying corollary 2 with $t = \frac{\lambda_{reg}}{10\alpha K}$ we get

$$\mathbf{P}\left(\sum_{i=1}^{m_k} \phi_i^{(k)} \phi_i^{(k),\top} \preccurlyeq \left(\frac{\lambda_{reg}}{\alpha K}I + 2m_k\, \mathbb{E}_{\phi\sim d^{(k)}}[\phi\phi^\top]\right)\right) \geq 1 - 2dM\exp\left(-\frac{\lambda_{reg}}{40\alpha K}\right). \quad (109)$$

Now multiplying the event inside the probability by $\alpha = \frac{n_k}{m_k} = \frac{N}{M}$ we get

$$\Sigma^{(k)} \stackrel{def}{=} \alpha \sum_{i=1}^{m_k} \phi_i^{(k)} \phi_i^{(k),\top} \preccurlyeq \frac{\lambda_{reg}}{K}I + 2n_k\, \mathbb{E}_{\phi\sim d^{(k)}}[\phi\phi^\top]. \quad (110)$$

After a union bound on the number of phases we can write

$$\mathbf{P}\left(\Sigma_M \stackrel{def}{=} \sum_{k=1}^{K} \Sigma^{(k)} + \lambda_{reg}I \preccurlyeq 2\lambda_{reg}I + 2\sum_{k=1}^{K} n_k\, \mathbb{E}_{\phi\sim d^{(k)}}[\phi\phi^\top] \stackrel{def}{=} 2\overline{\Sigma}\right) \quad (111)$$

$$\geq 1 - 2dMK\exp\left(-\frac{\lambda_{reg}}{40\alpha K}\right). \quad (112)$$

and substituting the value of $\alpha$ on the right hand side now gives

$$\mathbf{P}\left(\Sigma_M \preccurlyeq 3\overline{\Sigma}\right) \geq 1 - 2dMK\exp\left(-\frac{\lambda_{reg}M}{40KN}\right). \quad (113)$$

The final result follows from basic algebra. $\qquad\square$

**Lemma 15** (Matrix Upper Bound Online Phase)**.** *Recall that*

$$\Sigma'_n = \sum_{j=1}^{n-1} \phi(s'_j, a'_j)\phi(s'_j, a'_j)^\top + \lambda_{reg}I, \qquad \overline{\Sigma} = \alpha \sum_{i=1}^{M} \mathbb{E}_{\phi\sim d_i}\phi\phi^\top + \lambda_{reg}I. \quad (114)$$

*Algorithm 2 produces a cumulative covariance matrix $\Sigma'_N$ that satisfies*

$$\mathbf{P}\left(9\Sigma'_N \succcurlyeq \overline{\Sigma}\right) \geq 1 - \frac{\delta}{4} \quad (115)$$

*as long as $\lambda_{reg} \geq 24\ln\frac{8d}{\delta}$.*

*Proof.* Notice that conditioned on the run of algorithm 1, the distributions $d^{(1)}, \ldots, d^{(K)}$ of the features $\phi$ corresponding to the policies $\pi^{(1)}, \ldots, \pi^{(K)}$ are fixed (non-random), hence $\overline{\Sigma} = \sum_{k=1}^{K} \Sigma^{(k)}$ is non-random. Also, let $\phi'_i$ be the $i$ feature vector collected during the online phase. Notice that conditioned on all the random variables during the offline portion we can write

$$\mathbb{E}\,\Sigma'_N = \lambda_{reg}I + \mathbb{E}\sum_{i=1}^{N}(\phi'_i)(\phi'_i)^\top \quad (116)$$

$$= \lambda_{reg}I + \sum_{k=1}^{K} \mathbb{E}\sum_{i=1}^{n'_k} \mathbb{E}_{\phi\sim d^{(k)}}\phi\phi^\top \quad (117)$$

where $n'_k$ is the number of times that policy $\pi^{(k)}$ is sampled during the online phase. Continuing,

$$= \lambda_{reg}I + \sum_{k=1}^{K} \mathbb{E}\,n'_k\, \mathbb{E}_{\phi\sim d^{(k)}}\phi\phi^\top \quad (118)$$

$$= \lambda_{reg}I + \sum_{k=1}^{K} n_k\, \mathbb{E}_{\phi\sim d^{(k)}}\phi\phi^\top \quad (119)$$

$$\stackrel{def}{=} \overline{\Sigma}. \tag{120}$$

Now, apply Lemma 13 *(Matrix Lower Bound)* with $\lambda = \lambda_{reg}/(3N)$ we get

$$\mathbf{P}\left(8\left(3N\lambda I + \sum_{i=1}^{N}(\phi_i')(\phi_i')^\top\right) \succcurlyeq \sum_{k=1}^{K} n_k \, \mathbb{E}_{\phi \sim d^{(k)}} \, \phi\phi^\top\right) \tag{121}$$

$$\geq 1 - 2d\exp\left(-\frac{N\lambda}{8}\right) = 1 - 2d\exp\left(-\frac{\lambda_{reg}}{24}\right). \tag{122}$$

Recall that $\Sigma'_N = \lambda_{reg}I + \sum_{i=1}^{N}(\phi_i')(\phi_i')^\top$ and $\overline{\Sigma} = \lambda_{reg}I + \sum_{k=1}^{K} n_k \, \mathbb{E}_{\phi \sim d^{(k)}} \, \phi\phi^\top$. Eq. (121) implies that

$$\mathbf{P}\left(3N\lambda I + 8\left(\underbrace{\underbrace{3N\lambda}_{\lambda_{reg}}I + \sum_{i=1}^{N}(\phi_i')(\phi_i')^\top}_{\Sigma'_N}\right) \succcurlyeq \underbrace{3N\lambda I + \sum_{k=1}^{K} n_k \, \mathbb{E}_{\phi \sim d^{(k)}} \, \phi\phi^\top}_{\overline{\Sigma}}\right) \tag{123}$$

$$\geq 1 - 2d\exp\left(-\frac{\lambda_{reg}}{24}\right). \tag{124}$$

By setting $\lambda_{reg} \geq 24\ln\frac{8d}{\delta}$ we have

$$\mathbf{P}\left(9\Sigma'_N \succcurlyeq \overline{\Sigma}\right) \geq 1 - \delta/4. \tag{125}$$

$\square$

# F   Scalar Concentration Inequalities for Martingales

## F.1   Bernstein Inequality for Martingales

The following lemma is the same as Theorem 1 from Beygelzimer et al. (2011) as is reported here for completeness.

**Theorem 2** (Bernstein's Inequality for Martingales). *Consider the stochastic process $\{X_t\}$ adapted to the filtration $\{\mathcal{F}_t\}$. Assume $X_t \leq 1$ almost surely. Then*

$$\forall \lambda \in (0,1], \qquad \mathbf{P}\left(\sum_{t=1}^{T} X_t \leq \lambda \sum_{t=1}^{T} \mathbb{E}_t X_t^2 + \frac{1}{\lambda}\ln\frac{1}{\delta}\right) \geq 1 - \delta, \tag{126}$$

*which implies*

$$\mathbf{P}\left(\sum_{t=1}^{T} X_t \leq 2\sqrt{\left(\sum_{t=1}^{T} \mathbb{E}_t X_t^2\right)\ln\frac{1}{\delta} + 2\ln\frac{1}{\delta}}\right) \geq 1 - \delta. \tag{127}$$

For completeness, we reprove the theorem below:

*Proof.* Define the random variable $M_t$ as

$$M_t = M_{t-1}\exp(\lambda X_t - \lambda^2 \, \mathbb{E}_t X_t^2) \tag{128}$$

where in particular $M_0 = 1$. Recall the inequality $e^x \leq 1 + x + x^2$ for $x \leq 1$ and $1 + x \leq e^x$:

$$\mathbb{E}_t M_t = M_{t-1} \, \mathbb{E}_t \exp(\lambda X_t - \lambda^2 \, \mathbb{E}_t X_t^2) \tag{129}$$

$$\leq M_{t-1} \, \mathbb{E}_t\left[(1 + \lambda X_t + \lambda^2 X_t^2)\right]\exp(-\lambda^2 \, \mathbb{E}_t X_t^2) \tag{130}$$

$$\leq M_{t-1}\exp(\lambda^2 \, \mathbb{E}_t X_t^2)\exp(-\lambda^2 \, \mathbb{E}_t X_t^2) \tag{131}$$

$$= M_{t-1}. \tag{132}$$

Thus $\{M_t\}$ is a supermartingale sequence adapted to $\{\mathcal{F}_t\}$. In particular, $\mathbb{E}\, M_t \le M_0 = 1$ using the tower property. Now by the Markov inequality

$$\mathbf{P}\left(\underbrace{\lambda \sum_{t=1}^{T} X_t - \lambda^2 \sum_{t=1}^{T} \mathbb{E}_t\, X_t^2 > \ln \frac{1}{\delta}}_{\ln M_t}\right) = \mathbf{P}\left(M_t > \frac{1}{\delta}\right) \le \frac{\mathbb{E}\, M_t}{\frac{1}{\delta}} \le \delta \tag{133}$$

This implies that with probability at least $1 - \delta$ the following event holds:

$$\lambda \sum_{t=1}^{T} X_t - \lambda^2 \sum_{t=1}^{T} \mathbb{E}_t\, X_t^2 = \ln M_t \le \ln \frac{1}{\delta} \tag{134}$$

which is the first part of the thesis. Now, we choose $\lambda$. If $\sum_{t=1}^{T} \mathbb{E}_t\, X_t^2 \le \ln \frac{1}{\delta}$ then under the above event we obtain with $\lambda = 1$ (the largest possible value)

$$\sum_{t=1}^{T} X_t \le \sum_{t=1}^{T} \mathbb{E}_t\, X_t^2 + \ln \frac{1}{\delta} \le 2 \ln \frac{1}{\delta}. \tag{135}$$

If conversely $\sum_{t=1}^{T} \mathbb{E}_t\, X_t^2 \ge \ln \frac{1}{\delta}$ then let $\lambda = \sqrt{\frac{\ln \frac{1}{\delta}}{\sum_{t=1}^{T} \mathbb{E}_t\, X_t^2}} \le 1$ to obtain (still under the same event)

$$\sum_{t=1}^{T} X_t \le 2 \sqrt{\left(\sum_{t=1}^{T} \mathbb{E}_t\, X_t^2\right) \ln \frac{1}{\delta}}. \tag{136}$$

Therefore, summing the rhs of eqs. (135) and (136) to cover both cases we obtain the second part of the thesis. $\qquad\square$

## F.2   Reverse Bernstein Inequality for Martingales

Bernstein's inequality bounds the sum a random variable $\sum_t X_t$ using second moment information $\sum_t \mathrm{Var}_t\, X_t$; in our case (positive random variables in $[0, 1]$), the sum of the conditional variances $\sum_t \mathrm{Var}_t\, X_t$ is upper bounded by the sum of the means $\sum_t \mathbb{E}_t\, X_t$.

This section provides the 'reverse' inequality: Assuming a bound on the sum a random variable $\sum_t X_t$, it bounds the conditional sum of the means $\sum_t \mathbb{E}_t\, X_t$.

**Theorem 3** (Reverse Bernstein for Martingales). *Let $(\Sigma, \mathcal{F}, \mathbf{P})$ be a probability space and consider the stochastic process $\{X_t\}$ adapted to the filtration $\{\mathcal{F}_t\}$. Let $\mathbb{E}_t\, X_t \overset{def}{=} \mathbb{E}[X_t \mid \mathcal{F}_{t-1}]$ be the conditional expectation of $X_t$ given $\mathcal{F}_t$. If $0 \le X_t \le 1$ then it holds that*

$$\mathbf{P}\left(\sum_{t=1}^{T} \mathbb{E}_t\, X_t \ge \frac{1}{4}\left(c_1 + \sqrt{c_1^2 + 4\left(\sum_{t=1}^{T} X_t + c_2\right)}\right)^2\right) \le \delta, \qquad c_1 = 2\sqrt{\ln \frac{1}{\delta}}, \; c_2 = 2 \ln \frac{1}{\delta} \tag{137}$$

*Proof.* Consider the random 'noise'

$$\xi_t \overset{def}{=} \mathbb{E}_t\, X_t - X_t \tag{138}$$

which allows us to write

$$\sum_{t=1}^{T} \mathbb{E}_t\, X_t = \sum_{t}(\xi_t + X_t) \tag{139}$$

Then Theorem 2 *(Bernstein's Inequality for Martingales)* ensures the following high probability statement for appropriate $c_1 = 2\sqrt{\ln \frac{1}{\delta}}$, $c_2 = 2\ln\frac{1}{\delta}$:

$$\mathbf{P}\left(\sum_{t=1}^{T}\xi_t \leq c_1\sqrt{\sum_{t=1}^{T}\mathbb{E}_t\xi_t^2} + c_2\right) \geq 1 - \delta. \tag{140}$$

Notice that since $0 \leq X_t \leq 1$ we have

$$\mathbb{E}_t\,\xi_t^2 = \mathbb{E}_t(X_t - \mathbb{E}_t\,X_t)^2 \tag{141}$$
$$= \mathbb{E}_t\,X_t^2 - (\mathbb{E}_t\,X_t)^2 \tag{142}$$
$$\leq \mathbb{E}_t\,X_t^2 \tag{143}$$
$$\leq \mathbb{E}_t\,X_t. \tag{144}$$

Plugging back into the above display and using eq. (139) gives

$$\mathbf{P}\left(\sum_{t=1}^{T}\xi_t = \sum_{t=1}^{T}(\mathbb{E}_t\,X_t - X_t) \leq c_1\sqrt{\sum_{t=1}^{T}\mathbb{E}_t\,X_t} + c_2\right) \geq 1 - \delta \tag{145}$$

or equivalently

$$\mathbf{P}\left(\sum_{t=1}^{T}\mathbb{E}_t\,X_t \leq \sum_{t=1}^{T}X_t + c_1\sqrt{\sum_{t=1}^{T}\mathbb{E}_t\,X_t} + c_2\right) \geq 1 - \delta. \tag{146}$$

Solving for $\sum_{t=1}^{T}\mathbb{E}_t\,X_t$ gives under such event

$$\sum_{t=1}^{T}\mathbb{E}_t\,X_t \leq \frac{1}{4}\left(c_1 + \sqrt{c_1^2 + 4\left(\sum_{t=1}^{T}X_t + c_2\right)}\right)^2. \tag{147}$$

$\square$

## G  Additional Remarks

### G.1  Hard Instance that Requires Exploration

Even with a fixed context, uniform exploration ignores the structure of the reward function, and is therefore suboptimal. Consider the case where $d = 2$ and $\phi(1,1) = (1,0)$ and $\phi(1,i) = (0,1)$ for $2 \leq i \leq A$. The context is fixed and there are $A$ actions. A uniform exploration requires $\Omega(A/\epsilon^2)$ samples to estimate the reward for action 1. In contrast, the optimal exploration policy is

$$\pi(1) = \begin{cases} 1, & \text{w.p. } 1/2, \\ 2, & \text{w.p. } 1/2. \end{cases}$$

And the corresponding sample complexity is $\widetilde{O}(1/\epsilon^2)$.

On the other hand, the policy that ignores context is also suboptimal. Consider the policy $\pi^G(s)$ that returns the G-optimal design on the action set $\mathcal{A}_s$. The policy $\pi^G$ explores optimally for a fixed context. Ruan et al. (2020)'s Lemma 4 implies that this policy achieves an online sample complexity $\widetilde{O}(d^3/\epsilon^2)$. For completeness, we also include a hard instance for $\pi^G$. Let $\mathcal{S} = \{s_1, \cdots, s_k\}$ and $\mathcal{A}_s = \{a_1, \cdots, a_{k+1}\}$. Assume a uniform distribution over the state space $\mathcal{S}$. The feature vector is defined as follows.

$$\phi(s_i, a_j) = \begin{cases} e_j, & \text{when } j \leq k, \\ e_{i+k}, & \text{when } j = k+1. \end{cases} \tag{148}$$

Note that in this case, the dimension of the feature vector is $d = 2k$. For a fixed $s \in \mathcal{S}$, the G-optimal design returns the uniform exploration policy. As a result, the expected covariance matrix is

$$\Sigma = \mathbb{E}_{s \sim \mathcal{S}, a \sim \mathcal{A}}\, \phi(s,a)\phi(s,a)^\top = \mathrm{diag}(1/(k+1), \cdots, 1/(k+1), 1/k(k+1), \cdots, 1/k(k+1)).$$

It follows that

$$\mathbb{E}_{s \sim \mathcal{S}} \max_{a \in \mathcal{A}_s} \phi(s,a)^\top \Sigma^{-1} \phi(s,a) \geq \mathbb{E}_{s \sim \mathcal{S}}\, \phi(s,a_{k+1})^\top \Sigma^{-1} \phi(s,a_{k+1}) = k(k+1) = \Omega(d^2).$$

In contrast, the optimal exploration policy is

$$\pi(s_i) = \begin{cases} U(\{a_1, \cdots, a_k\}), & \text{w.p. } 1/2, \\ a_{k+1}, & \text{w.p. } 1/2, \end{cases} \tag{149}$$

where $U(\cdot)$ denotes the uniform distribution. Correspondingly, we have $\Sigma = \mathrm{diag}(1/2k, \cdots, 1/2k)$, and

$$\mathbb{E}_{s \sim \mathcal{S}} \max_{a \in \mathcal{A}_s} \phi(s,a)^\top \Sigma^{-1} \phi(s,a) \leq O(d). \tag{150}$$

### G.2 Remarks Regarding Ruan et al. (2020)

Section 6 of Ruan et al. (2020) argues that when $\lambda_{reg} < 1/d$, the covariance matrix doesn't concentrate. Their construction works as follows. Consider a fixed offline dataset $\mathcal{C}$ with size $M$. The context space is $\mathcal{S} = [d]$. The action space is $\mathcal{A}_1 = \{1\}$ and $\mathcal{A}_s = \{1, 2\}$ for $2 \leq s \leq d$. The feature vector is defined as

$$\phi(1,1) = e_1, \quad \phi(s,1) = e_s, \quad \phi(s,2) = \sqrt{1 - \frac{d}{M}} e_s + \sqrt{\frac{d}{M}} e_1. \tag{151}$$

The context distribution $\mu$ is

$$\mu(1) = \frac{1}{dM}, \quad \mu(s) = \frac{1}{d-1}\left(1 - \frac{1}{dM}\right), \forall s \geq 2. \tag{152}$$

Then with probability at least $1/d$, $\mathcal{C}$ contains a single occurrence of context 1. Ruan et al. (2020) argues that in this case, there exits a policy $\pi$ such that the covariance on $\mathcal{C}$ deviates from the population one when $\lambda_{reg} < \frac{1}{d}$.

The policy is $\pi(s) = 1$. Let $\hat{\Sigma} = \lambda_{reg} I + \sum_{s_i \in \mathcal{C}} \phi(s_i, \pi(s_i))\phi(s_i, \pi(s_i))^\top$. We can compute that

$$\mathbb{E}_{s \sim \mathcal{C}} \max_{a \in \mathcal{A}_s} \phi(s,a)^\top \hat{\Sigma}^{-1} \phi(s,a) \leq O(d/M). \tag{153}$$

Now consider the true distribution $\mu$. Let $\Sigma = \lambda_{reg} I + M\, \mathbb{E}_{s \sim \mu}\, \phi(s, \pi(s))\phi(s, \pi(s))^\top$. By basic algebra we get $\Sigma = \lambda_{reg} I + M \mathrm{diag}(1/(dM), \mu(2), \cdots, \mu(d))$. As a result, we can compute

$$\mathbb{E}_{s \sim \mu} \max_{a \in \mathcal{A}_s} \phi(s,a)^\top \Sigma^{-1} \phi(s,a)$$

$$= \mu(1)\frac{d}{2} + \sum_{s=2}^{d} \mu(s) \max\left(\frac{1}{\lambda_{reg} + M\mu(s)}, \left(1 - \frac{d}{M}\right)\frac{1}{\lambda_{reg} + M\mu(s)} + \frac{d}{M}\frac{1}{\lambda_{reg} + 1/d}\right)$$

$$\geq \Omega(d^2/M).$$

Note the inequality follows due to the rightmost term and substituting in $\lambda_{reg} < \frac{1}{d}$. Comparing the estimate of the empirical covariance with the true expectation, we observe the covariance matrix doesn't concentrate. Indeed, in the setting where $\lambda_{reg} < 1$, the concentration events $\mathcal{E}_1, \mathcal{E}_2, \mathcal{E}_3$ in the proof of Theorem 1 fail with constant probability. Since Ruan et al. (2020) focus on small regularization setting, their algorithm first finds a "core" of the contexts. This procedure makes their algorithm much more complex compared with ours, and increases the number of offline samples required.

Now if we set $\lambda_{reg} = 1$, by the same computation we get

$$\mathbb{E}_{s \sim \mu} \max_{a \in \mathcal{A}_s} \phi(s,a)^\top \Sigma^{-1} \phi(s,a) \leq \Omega(d/M).$$

Hence, the concentration events hold and Theorem 1 gives a much tighter offline sample complexity.

# H  Helper Lemmas

**Lemma 16** (Number of Switches). *Algorithm 1 generates at most $K$ distinct policies:*

$$K \leq d \ln_2 \left( 1 + \frac{M}{d\lambda_{reg}} \right) = \widetilde{O}(d) \tag{154}$$

*Proof.* Notice that $\det(\Sigma_1) = \lambda_{reg}^d$ and $\det(\Sigma_M) \leq (\lambda + \frac{M}{d})^d$ (see proof of lemma 11 in (Abbasi-Yadkori et al., 2011)). Every time the policy changes the determinant of $\Sigma_m$ at least doubles. Let $K$ denote the number of times the policy is updated. By induction,

$$\left( \lambda_{reg} + \frac{M}{d} \right)^d \geq \det(\Sigma_M) \geq 2^K \det(\Sigma_1) \geq 2^K \lambda_{reg}^d \tag{155}$$

Solving for $K$ concludes.  □

# I  Additional Experiments and Information

## I.1  Synthetic Dataset

Here we describe how the covariance matrices were defined for the synthetic experiment based on the categories and actions. Recall that this applies for action $a \in \{1, 2, 3\}$ and category $i \in \{1, 2, 3\}$. Recall also that all covariance matrices $\Sigma_{a,i}$ are diagonal of the form $\mathrm{diag}(10^{-9}, \ldots, 10^{-9}, 1, 10^{-9}, \ldots, 10^{-9})$ an differ only by the placement of the coordinate that is equal to 1.

1. For category $i = 1$: $(\Sigma_{1,1})_{11} = 1$, $(\Sigma_{2,1})_{22} = 1$, $(\Sigma_{3,1})_{33} = 1$.
2. For category $i = 2$: $(\Sigma_{1,2})_{44} = 1$, $(\Sigma_{2,2})_{11} = 1$, $(\Sigma_{3,2})_{55} = 1$.
3. For category $i = 3$: $(\Sigma_{1,3})_{66} = 1$, $(\Sigma_{2,3})_{77} = 1$, $(\Sigma_{3,3})_{11} = 1$.

## I.2  Yahoo! Learning to Rank Dataset

The Yahoo! dataset is available freely for research purposes through Yahoo! Webscope. Use of the dataset required accepting an agreement not to share the original in a way that the dataset could be reconstructed. The dataset is available at the following link:

https://webscope.sandbox.yahoo.com/catalog.php?datatype=c

The data consists entirely of numerical feature vectors and does not contain any identifiable or offensive information. To run the experiments, we used a standard Amazon Web Services EC2 c5.xlarge instance with 4 vCPUs and 8gb of memory.

For the subsampling to create 300-dimensional feature vectors, we selected coordinates randomly by sampling with replacement to include out of the full 700.

## I.3  Additional Plots

In addition to the three choices of regularization shown in Figure 3 for the real-world dataset, we ran additional more extreme values, but omitted them for clarity in the plot. In Figure 4, we see the same algorithms plotted with both large values of regularization ($\geq 1$) and small values of regularization ($\leq 1$).

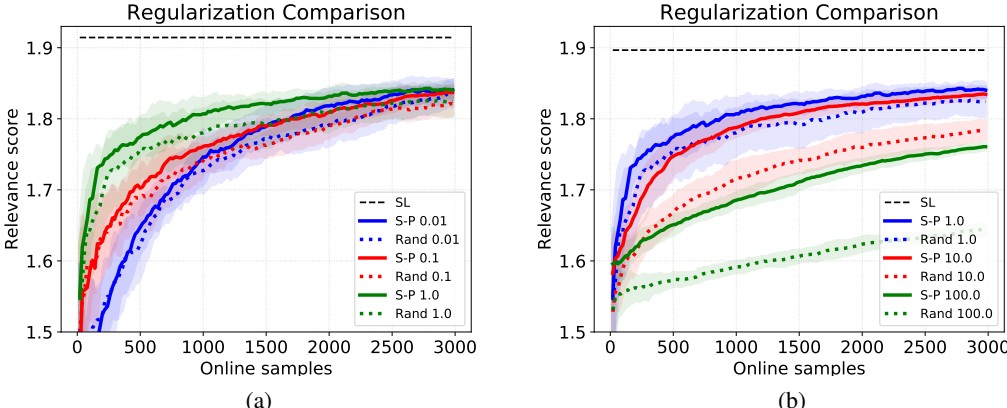

Figure 4: The left figure shows the sampler-planner (S-P) compared to the random algorithm with small regularization $\lambda \in \{0.01, 0.1, 1.0\}$. The right shows the same for $\lambda \in \{1.0, 10.0, 100.0\}$.