# OpenReview forum: "Design of Experiments for Stochastic Contextual Linear Bandits"
_NeurIPS.cc/2021/Conference — NeurIPS 2021 Poster_

### Official Review · Reviewer_tDzP · 2021-07-06

**Rating:** 6
**Confidence:** 3

**Summary:**

The authors propose a new setting for the stochastic contextual linear bandit problem: using past contexts only, design a non-reactive policy for future online data. They argue that this setting arises in practice since often it may be logistically too complicated to deploy an online machine learning algorithm. They measure the quality of the dataset by the performance of the greedy policy that uses the least-square predictor on the collected data in the online phase. They develop a new two-phase algorithm (a planner and a sampler) for this problem and provide sample complexity guarantees. They also demonstrate the performance of their algorithm empirically.

**Limitations And Societal Impact:**

Yes. They discuss using an algorithm that is robust to all values of regularization and the issue of the offline and online distributions potentially differing.

**Main Review:**

I think that the setting is original and is convincingly related to practice and I consider it a contribution in and of itself. The algorithm is interesting, especially the application of reward-free LinUCB in the planner phase. Section 6 is helpful and clarifies the main steps in the argument. The experiments demonstrate the usefulness of the algorithm.

The authors claim that the algorithm achieves the minimax lower bound on the number of online samples, but do not state a bound or give a reference for this. This would be helpful to clarify. It is also not clear whether the number of offline contexts is optimal. Can something be said about this? A discussion would be helpful.

It seems that while the planner never uses either the rewards or the actions available in the offline. A discussion might be helpful highlighting this. Is there any reason to believe that these could be used in an algorithm to achieve improved performance? In practice, does one ever have only the contexts and not the rewards or actions?

After discussion: the algorithm seems useful, but the novelty of the setting and the technical contribution seem more limited than I originally thought. I think a detailed discussion of the connection to the reward-free RL literature would be helpful.

**Time Spent Reviewing:**

4

---

> ### Author Response · Authors · 2021-08-10
> **Detailed answer to reviewer's comments**
>
> We thank the reviewer for the positive review.
>
> 1) ''the algorithm achieves the minimax lower bound on the number of online samples, but do not state a bound or give a reference for this'’
>
> Thank you for raising this: we completely agree that describing this in more detail will be a valuable addition. Indeed, the precise minimax lower bound result in terms of sample complexity is not explicitly stated in the literature; most lower bounds are derived for the more  challenging regret setting, but can nonetheless be  adapted to derive sample complexity results.
>
> For the large action regime (i.e., $\min (\ln |\mathcal S \times \mathcal A |,d ) = d$), the lower bound $\approx d^2/\epsilon^2$ follows by using the same hypercube construction as Theorem 24.1 of the book ''Bandit Algorithms’’ by Lattimore and Szepesvari, 2020.
>
> For the small action regime (i.e., $\min (\ln |\mathcal S \times \mathcal A |,d ) = \ln |\mathcal S \times \mathcal A |$). If $O(|A|) =  d$, and we ignore the $\ln |\mathcal S \times \mathcal A |$ term in sample complexity, the lower bound $\approx d/\epsilon^2$ follows  by considering a bandit instance with $(d)$ orthogonal arms in $\mathbb R^d$.
> This is equivalent to a multi armed bandit problem, and the sample complexity lower bound then follows from Theorem 1 in ''The Sample Complexity of Exploration in the Multi-Armed Bandit Problem’’ by Mannor and Tsitsiklis, 2004.
>
> We will update the text to include the above.
>
> 2) ''It is also not clear whether the number of offline contexts is optimal. Can something be said about this?’’
>
> Indeed this is an open question, and we do not know the answer to this. We conjecture that it is improvable, but new concentration inequalities on concentration of covariances might be needed to carefully answer this question and derive a new algorithm. We will include a short statement about this.
>
> 3) ''It seems that while the planner never uses either the rewards or the actions available in the offline. A discussion might be helpful highlighting this. Is there any reason to believe that these could be used in an algorithm to achieve improved performance? ’’
>
> We completely agree this is an important and interesting issue and we will add a short discussion as suggested.  We expect this will improve performance since if some reward data is already available, the planner may be able to focus on less explored regions of the action space, potentially pruning away certain parts of the action space if their performance is known to be dominated by other actions in certain parts of the context space, or
> possibly even reducing the dimensionality $d$. We believe this is a very interesting issue for future work.
>
> 4) ''In practice, does one ever have only the contexts and not the rewards or actions?''
>
> Thanks for allowing us to clarify this point. In many settings there is unsupervised data about state contexts-- such as features about customers. Our approach allows us to leverage such data to design an exploratory policy. For example, a social media company may have a lot of information about its existing users before it launches a new personalized news recommendation engine for those users, even though it has no data yet about the rewards of providing these new actions (news articles) to its users.

---

### Official Review · Reviewer_SJn5 · 2021-07-09

**Rating:** 5
**Confidence:** 4

**Summary:**

This paper studies the experimental design for the contextual linear bandits. The goal of the problem is to collect a dataset for identifying a good policy (an estimation of the reward vector). This paper provides algorithms for their goal and theoretically analyzes the sample complexity. Experimental results are also provided.

**Limitations And Societal Impact:**

See section 8.

**Main Review:**

The paper is well-written. Although I didn't carefully check all the proofs, the theoretical results intuitively make sense to me.

The problem can be viewed as a reward-free exploration in linear contextual bandits. My main concern is regarding the significance of the problem studied, given existing work on reward-free RL, i.e., see Jin et al 2020, "Reward-Free Exploration for Reinforcement Learning" and many follow-up works (some of them also study the case with linear function approximation, e.g., Wang et al 2020, "On reward-free reinforcement learning with linear function approximation"). Since contextual bandit can be viewed as a special case of RL by setting horizon H=1, I wonder if the results presented in this paper can be directly obtained from these existing results. Discussions/comparisons with respect to reward-free RL are missing in the paper.

Another question is with respect to Ruan et al 2020. The author mentioned that similar results can be derived from their Thm 6 and offline complexity O(d^{16}). I agree that d^16 is high-suboptimal. However, it seems that the more important parameter is 1/\epsilon. Their results completely get rid of dependence on 1/\epsilon. Results derived in this paper have 1/\epsilon^2 dependence, which looks suboptimal to me.

====after rebuttal====

I have read the rebuttal and would like to keep my score at 5. I suggest a major revision of the current submission to include a detailed discussion of the reward-free RL literature, e.g., what can be directly obtained from existing results and what are the new results in the current submission. Personally, I would not take the problem setting as a main/novel contribution of this paper due to its similarity to existing papers.

**Time Spent Reviewing:**

3

---

> ### Author Response · Authors · 2021-08-10
> **Our setting differs from reward free-exploration**
>
> We thank the reviewer for the constructive feedback.
>
> 1) “reward-free exploration in linear contextual bandits [...]. I wonder if the results presented in this paper can be directly obtained from these existing results”
>
> We apologize for not putting more emphasis on the distinction with the reward free exploration setting; in short, reward-free exploration does *not* cover our setting. This is because the reward free exploration literature uses reactive/adaptive  algorithms, i.e., algorithms that  change their policy as they explore the environment in a reward-agnostic way. This differs from our setting, where the goal is to prescribe a *single* exploratory policy. This is motivated by settings where changing the policy is expensive or logistically challenging (consider if every policy update requires manual intervention by a human). For example, consider public health  organizations sending text messages to encourage vaccines. While text message systems often enable user segmentation to enable  personalization, such organizations may be not yet willing to invest in the costs needed to implement a continuously improving contextual MAB system. However, deploying a single exploratory policy is similar to running a AB test, but can yield the information needed to deploy a context-specific policy in the future.
>
>
> 2) Ruan et al 2020 $1/\epsilon$ dependence
>
> We agree both settings are important, and here our focus was on algorithms for practical settings where
> the policy class can be large (i.e., large $d$). In such settings, Ruan et al's $d^{16}$ dependence will be far worse than our $d^2/\epsilon^2$ dependence for most $\epsilon$. Our work compliments their prior literature by focusing on this other important setting, and we will be sure to update the text to clarify this point.

---

> > ### Comment · Reviewer_SJn5 · 2021-08-27
> > **response to authors**
> >
> > Thank you for your reply. However, I'm a little skeptical about the claim ''... in short, reward-free exploration does not cover our setting. This is because the reward free exploration literature uses reactive/adaptive algorithms ...''
> >
> > It seems to me that Algorithm 2 in ''Reward-Free Exploration for Reinforcement Learning'' by Jin et al. 2020 can be decomposed into two parts: (1) line 3-7 corresponds to the ''planner'' in this submission; and (2) line 8-11 corresponds to the ''sampler'' in the submission (they are basically the same). Line 3-7 is reactive, however, the same holds true for the ''planner'' in this submission: it uses offline data points. I strongly suspect that lines 3-7 in Alg2 by Jin et al. 2020 can be modified so that it only requires offline data (at least for the simpler bandit setting without planning). That being said, I'm not convinced that the setting studied in this paper is novel in itself.

---

> > > ### Author Response · Authors · 2021-08-28
> > > **Differences in the analysis between reactive and non-reactive algorithms**
> > >
> > > Thank you for highlighting the quite similar two-stage structure of the tabular MDP reward free algorithm of Jin et al and also for the opportunity to more precisely explain the important differences between our work and the existing reward-free papers for MDPs.
> > >
> > > We completely agree that the structure of Jin et al’s algorithm is similar (though it uses all online samples) and could likely be adapted to when the planner uses offline data. We will update the text to include this interesting connection. Jin et al. operates in the tabular case with no structure in the state-action space. Note for simple contextual multi-armed bandits (tabular states, no shared structure, Jin’s setting with H=1), a good strategy in the non-adaptive setting is to simply sample actions uniformly; this is optimal given the lack of shared information between state-actions. No sophisticated algorithm is needed in that case.
> > >
> > > In the linear bandit setting the closest work we are aware of is Wang et al “On Reward-Free Reinforcement Learning with Linear Function Approximation.” While their algorithm can be specialized to the linear bandit setting and modified to match our planner-sampler algorithm by using offline data, the theoretical analysis would need to be very different from that in Wang et al.’s paper (and thus it would not yield the results presented in our paper). Indeed, when decoupling the planner from the sampler (to identify a non-reactive policy), the planner needs to accurately predict what will happen when the sampler is run. To ensure this, a key challenge is to ensure that the planner inverse covariance matrix is very similar to the sampler inverse covariance matrix. This issue does not arise in Wang et al’s algorithm as they can avoid the planner-sampler structure since they focus on reactive policies. This is a key distinction which is highlighted in line 239-244 in our submission; a significant portion of the appendix is devoted to establishing such claim. This also reveals a potential tradeoff between different levels of regularization and number of offline / online samples for the setting we consider (in general, more regularization and more offline samples help the inverse covariance matrix estimation).
> > >
> > > We will update the text to include the above discussion.

---

### Official Review · Reviewer_mozb · 2021-07-12

**Rating:** 6
**Confidence:** 4

**Summary:**

[summary]

This paper considers the problem of efficient nonadaptive sampling to find the best bandit policy, given an offline context data is available. The authors propose an algorithm and provide its sample complexity analysis. Empirical results show the efficicay of the proposed method.


**Ethical Concerns:**

No concerns.

**Limitations And Societal Impact:**

Yes.

**Main Review:**



Originality 4/5: I think the problem is novel.

Quality 4/5: It's fine for the first paper on this problem.

Clarity 3/5: There are some typos in important places, but I am sure they are addressable.

Significance 3/5: this is hard to say, but I don't see an immediate clear application (see below for my comments)

Overall, the problem is novel, but I'd like to see a more specific scenarios where it make sense. Otherwise, the contribution seems solid as a first solution on this problem, modulo some details as I describe below.

I'd like to see a former argument on the lower bound. It's not immediately obvious how Chu'11 and Abbasi-Yadkori'11 would lead to the stated bound in line 179 (although I am convinced that the stated rate should be the right one).

I'd like to see some discussions on the tightness of M >= d^3/\eps^2 for small $S\times A$. Is it a technical difficulty or is it unimprovable?

Please identify the actual application of the proposed problem. The proposed problem is not reward maximization but policy identification, so I don't think one would like to use it for recommendation systems becuase it will not maximize the cumulative reward. It will have to be like $\epsilon$ greedy where one uses this sampler for $\epsilon$ portion of the traffic. Otherwise, there are pure exploration problems like A/B testing.

re: Eq (5).  consider the arm set consists of canonical vectors (e_1, ..., e_d) and when we pull arm $e_i$ like $N_i$ times and use the MLE to estimate $\hat\theta$. Consider that the context space is a singleton, so that $|S|\times|A|=d$. Then, we have
$$\\|\hat\theta-\theta^*\\|^2_{\Sigma} = \sum_{i=1}^d N_i  (\hat{\theta}_i - \theta^\*_i)^2  \lessapprox \sum_i N_i \frac{1}{N_i} = d$$
, which I believe is not improvable. This seems to conflict Eq. (5) which implies that we should get $\log(d)$ instead of $d$.

* reading the proof in later sections, I realized that perhaps the LHS of Eq (5) is a typo, and in the derivation one should not apply the cauchy-schwarz in Eq. (3) & (4).

The description of $\alpha$ is confusing because it was not explicitly defined in the main text (my apologies if I missed it). Although it was impilcitly mentioned in line 163, L155 says 'a key choice is the parameter $\alpha$...', which will confuse readers.  Theorem 1 should also explicitly state what  $\alpha$ one has to use.

[minor comments]

* why say 'nonreactive' rather than 'nonadaptive', which is the standard terminology..?
* I would increase the resolution of Figure 1 for better readability.
* L199: \Sigma_M => \Sigma_m??
* L203: remove prime from U'_m
* Proof of Theorem 1
  * Eq (15): why do we have \log(N) instead of \sqrt{\log(N)}??
* experiments
  * clarification: for each trial, the arms are drawn and then fixed throughout, right?
  * L269-271: explanation here can be improved. It starts by 'in category i', but then it uses \Sigma_1 in the normal distribution. I would assume that it was meant to be \Sigma_i.
  * please define `policy value' mathematically.
* figure 2: (b) it sems like S-P would pull arm 6 and 7 for some time, but the description from the main text (L272) indicate that they should be 5 and 6. There seems to be a bug/typo here?

----
**after rebuttal**
I am satisfied with the rebuttal. I will keep the score.

**Time Spent Reviewing:**

5

---

> ### Author Response · Authors · 2021-08-10
> **Detailed answer to reviewer's comments**
>
> We thank the reviewer for the positive and thoughtful review.
>
> 1) ''It’s not immediately obvious how Chu’11 and Abbasi-Yadkori’11 would lead to the stated bound in line 179’’
>
> We completely agree this is describing this explicitly will improve the paper.
>
> -> For the sample complexity upper bounds: Abbasi-Yadkori’s algorithm achieves a $\tilde{O}(d^2/\epsilon^2)$ upper bound as follows.
> Their algorithm achieves a regret after $N$ timesteps of at most $d\sqrt{N}$, meaning that $\sum_{t=1}^{N}(r(s_t,\pi^\star(s_t))-r(s_t,\pi_t(s_t)))\le O(d\sqrt{N})$.
> Note that $\pi_t$ is independent of $s_t$, by the Azuma martingale concentration bound we get $\sum_{t=1}^{N}(E_{s}r(s,\pi^\star(s)) - E_{s}r(s,\pi_t(s)))\le O(d\sqrt{N}).$
>
> Consider the average policy $\pi(s)=\pi_t(s)\text{ w.p. }1/N$. We get $\mathbb{E}_\pi\mathbb{E}_s[r(s,\pi(s))]\ge \mathbb{E}_s[r(s,\pi^\star(s))]-O(d/\sqrt{N}).$ So when $N \geq \Omega(\frac{d^2}{\epsilon^2})$ the average policy played by  Abbasi-Yadkori’s algorithm from timestep $1$ to $N$ is at most $\epsilon$-suboptimal.
>
> -> For the sample complexity lower bounds: indeed, the precise minimax lower bound result in terms of sample complexity is not explicitly stated in the literature; most lower bounds are derived for the more  challenging regret setting, but can nonetheless be  adapted to derive sample complexity results.
>
> For the large action regime (i.e., $\min \{\ln |\mathcal S \times \mathcal A|,d \} = d$), the lower bound $\approx d^2/\epsilon^2$ follows by using the same hypercube construction as Theorem 24.1 of the book ``Bandit Algorithms’’ by Lattimore and Szepesvari, 2020.
>
> For the small action regime (i.e., $\min \{\ln |\\mathcal S \times \mathcal A |,d \} = \ln |\mathcal S\times \mathcal A |$). If $O(|A|) =  d$, and we ignore the $\ln |\mathcal S\times \mathcal A|$ term in sample complexity, the lower bound $\approx d/\epsilon^2$ follows  by considering a bandit instance with $(d)$ orthogonal arms in $\mathbb R^d$.
> This is equivalent to a multi armed bandit problem, and the sample complexity lower bound then follows from Theorem 1 in ``The Sample Complexity of Exploration in the Multi-Armed Bandit Problem’’ by Mannor and Tsitsiklis, 2004.
>
> We will include this in the appendix and reference these results in the main text.
>
> 2) ''tightness of $M \geq d^3/\epsilon^2$ for small $\mathcal S \times \mathcal A$’’
>
> This is an interesting issue. We do not yet know what is the optimal scaling for the number of offline samples. We think the result is improvable, but doing so would likely require new tools to analyze the concentration of covariance matrices, which is needed to reason about the `progress’ of the algorithm. We will add this short discussion to the text.
>
> 3) ''Please identify the actual application of the proposed problem’’
>
> There increasingly exist settings which lack the infrastructure for continuous online contextual multi-armed bandit learning, but that can implement (and would benefit from) contextualized decision policies. For example, educational startups, public policy groups and governmental agencies can use email and text messages to provide targeted information and opportunities, such as information to encourage vaccination or tips to parents to support their child's stage of development. Such organizations are familiar with standard experimental design, and the design of experiments work we propose could help fit in with standard workflows but enable the data-efficient identification of contextualized policies instead of AB testing or relying on other user segmentation methods that may not directly optimize the outcome of interest. We will be sure to clarify this in the text.
>
> 4) ''reading the proof in later sections, I realized that perhaps the LHS of Eq (5) is a typo, and in the derivation one should not apply Cauchy-Schwarz in Eq. (5)’’
>
> We thank the reviewer for spotting this! In the explanation in the main text (as the reviewer suggests) one should not go ahead and use Cauchy-Schwartz for the small action case in eq. (3) and (4); instead, confidence intervals should be written for every single arms + union bounds as done in the appendix. We will correct this.
>
> 5) ''description of $\alpha$ is confusing’’
>
> We apologize for not more clearly describing $\alpha$: it is the ratio of offline / online samples and we will be sure to update the text to make this clearer.
>
> 6) Minor comments:
> - Nonreactive vs nonadaptive: we are happy to make this change.
> - Thank you for identifying the typos on L199 and L203 and the suggestion to increase the resolution of Figure 1
>
> Experiments: Thanks for identifying unclear parts in the experiments. We will make the suggested changes. Below are answers to individual questions.
>
> - ''clarification: for each trial, the arms are drawn and then fixed throughout, right?''
>
> The arms are fixed over all trials and thus the distribution of features for a particular arm are fixed. States $s$ are drawn i.i.d at each time step.
>
> - ''L269-271: explanation here can be improved. It starts by 'in category i', but then it uses $\Sigma_1$ in the normal distribution. I would assume that it was meant to be $\Sigma_i$.''
>
> Yes, it should be $\Sigma_i$.
>
> - ''please define `policy value' mathematically.''
>
> Policy value is defined as $\mathbb E_{s \sim \mu } \left[  \phi(s, \hat \pi(s))^\top \theta^* \right]$, where $\hat \pi$ is defined in Equation 1 with the data learned so far. This quantity is estimated from samples from a held-out test set.
>
> - ''figure 2: (b) it sems like S-P would pull arm 6 and 7 for some time, but the description from the main text (L272) indicate that they should be 5 and 6. There seems to be a bug/typo here?''
>
> Yes, this is a typo. The text should read actions $6$ and $7$. The plot is correct.

---

### Official Review · Reviewer_UR5r · 2021-07-16

**Rating:** 6
**Confidence:** 2

**Summary:**

This paper studies the setting where it is not possible (e.g. not allowed or difficult) to update a policy while it interacts with a system. The authors suggest to build, from a batch of historical contexts, an explorative policy, non-reactive, that is not updated online as it interacts with the system, to gather informative data that are finally used to build a near-optimal policy.

The explorative policy (Planner) consists in a mixture of policies built offline from the historical contexts so as to minimize the uncertainty. The main idea is that using this mixture of policies on the real system will also gather data so as to reduce the uncertainty.

Sample complexity bounds and bounds on the performance of the final policy are derived. The algorithm is also compared with other strategies in numerical experiments.




**Limitations And Societal Impact:**

-

**Main Review:**

I think the setting studied by the authors is motivated by applications of policies in real life problems and such studies are missing from the literature. Therefore this could be a relevant contribution to the community.

The paper is well-written. Although the main ideas and algorithms are clear, some parts would benefit from more explanations, clarifications.

Major comments:
- please clarify the objective in the introduction: minimize the number of samples required to achieve a tolerance error of $\varepsilon$?
- the authors say that they apply LinUCB with an empirical reward function set to 0. Although I understand the algorithm, I am not sure to understand the part about setting the empirical reward function to 0.
- I would like the authors to elaborate on the necessity of using a mixture of policies.
-line 167 "in the presence of reward signal": I am confused because the reward is not used here by the fixed policy, it is just stored to build $\hat \pi$. I think the important thing is that the policy is fixed and non-reactive. It could be reactive by using the contexts only. The sentence used by the authors appear to be misleading here.
- Section 6: lots of typos, this section should be proof read. examples: through (line 201), at different speeds (line 211), missing or added $'$ at wrong places  (equation (9) and line 205), Lemma 2 (line 230).
- Experiments: can the author define "policy value"? The description of the synthetic dataset (lines 265-274) is hard to follow/understand. When the authors write "In category $i$, the action $a=i$", does $i$ belong to $\{1, 2, 3\}$. In Figure 2.(b), action 6 and 7 appear important while from the explanations and the setup it seems to me it should be action 5 and 6. For the learning to rank experiments, except for S-P 10 the result does not appear to be significant compared to the random strategy given the error bars. How are the error bars defined?
- Although the action space is state dependent $\mathcal{A}_s$ in the introduction, is it assumed in the bounds that we in fact have a product $\mathcal{S} \times \mathcal{A}$?
- How does the bounds/results compare with a strategy that would be able to update a policy online? the reactive policy could use the rewards and thus explore and reduce uncertainty where it is potentially useful for the reward. Whereas here we explore so as to reduce a global uncertainty not in the important directions? I think this is briefly mentionned in the conclusion, where such reward information could be used additionally to the contexts.

Minor comments:
- There is an issue with the biblio. 2 papers appear twice in the references. The issue is coming from "Szepesvári" vs "Szepesvari"
- line 35: a context-dependent action set $a \in \mathcal{A}_s$ -> although understandable this is not completely rigorous to put the $a$ here as you introduce a "set" in the text. please reformulate.
- Introduction: $d$ and $\epsilon$ are used at several places but are not defined.
- section 3. please define $\Omega$ notation.
- line 134: please explain the uncertainty (related to the G criterion from optimal design literature: minimizes the worst possible predicted variance).
- using both $\pi_e$ and $\pi_{mix}$ is confusing.
- Theorem 1: $\varepsilon$ should be defined at the beginning of the statement ("Let $\varepsilon \leq 1$) as it is used in $M$ and $N$.

**Time Spent Reviewing:**

4

---

> ### Author Response · Authors · 2021-08-10
> **Detailed answer to the reviewer's comments**
>
>
> Thank you for the thoughtful comments and questions. We will modify the text to address all the minor comments and here answer the major comments/questions.
>
> 1) ''objective: minimize the number of samples required to achieve a tolerance error of $\epsilon$”?
>
> Exactly: given a dataset of prior contexts, for a fixed failure probability $\delta$ and target accuracy $\epsilon$, our objective is to compute an efficient fixed exploratory policy (that is non-adaptive to the observed contexts) that can be used to gather online data that will enable us to return an $\epsilon$-optimal policy in expectation over the contexts. We wish to design the exploratory policy so as to minimize the online data samples needed.
> As a side goal, we want the algorithm to work even given only a small number of offline contexts.
>
> 2) ''the authors say that they apply LinUCB with an empirical reward function set to 0. Although I understand the algorithm, I am not sure to understand the part about setting the empirical reward function to 0.’’
>
> We are happy to clarify this in the text. If  LinUCB is executed with an empirical reward set to 0, the exploration bonus used in LinUCB will result in selecting in a state $s$ the action where $\| \phi(s,a) \|_{\Sigma^{-1}_m}$ is highest.
> In our approach, we compute our exploratory policy by running this reward free LinUCB on our historical contexts (note this can be done since we assume we can compute $\phi(s,a)$ for any state in our historical dataset and any potential action $a$).
>
> This way, our planner finds a
> sequence of policies which produce a covariance matrix $\Sigma_M$ such that upon termination $E_{s \sim \mu} \max_{a \in \mathcal A_s}\| \phi(s,a) \|_{\Sigma^{-1}_M}$ is small.
>
> 3) ''elaborate on the necessity of using a mixture of policies.''
>
> Thanks for the question: mixture policies are not necessary per se, they just arise naturally from the algorithm design since our planner finds a sequence of policies to produce a certain covariance matrix.
>
> 4) ''line 167 “in the presence of reward signal”: I am confused because the reward is not used here by the fixed policy, it is just stored to build $\widehat \pi$. I think the important thing is that the policy is fixed and non-reactive. It could be reactive by using the contexts only. The sentence used by the authors appear to be misleading here.’’
>
> The interpretation of the reviewer is correct, and we apologize for the confusion. The comment in the paper meant the following: the planner and the sampler are implementing the same mixture policy (at least when $\alpha = 1$), which suggests that both should create a similar cumulative covariance matrix $\Sigma$ up to sampling noise from the context distribution. Since $\Sigma$ effectively represents the amount of `information’ acquired, we can use the planner’s $\Sigma$ as a proxy for the sampler’s $\Sigma$. We will revise the sentence for additional clarity.
>
> 5) Section 6: typos
>
> Thanks for this and we will fix these.
>
> 6) Experiments:
>
> Thanks for pointing out unclear aspects of the experiments. We will update the text with these suggestions and clarifications.
>
> Policy value is defined as $\mathbb E_{s \sim \mu } \left[  \phi(s, \hat \pi(s))^\top \theta^* \right]$, where $\hat \pi$ is defined in Equation 1 with the data learned so far. This quantity is estimated from samples from a held-out test set.
>
> Yes, that is correct, $i \in \{1, 2, 3\}$ and, indeed, the text should read $a = 6, 7$ to denote the special "large norm" actions rather than $a = 5, 6$. We will also revise this paragraph to provide more intuition for the design and algorithm choices.
>
> As mentioned in the text, error bands represent standard deviation over the 10 independent trials. Consequently, they should not shrink (up to any estimation error).  If one were to use standard error this would be smaller by roughly a factor of $0.32$. As the reviewer points out, the difference is small in some; however, it is consistent.
>
> Furthermore, one typically does not know a good choice of $\lambda$ in online settings and Figure 3, along with figures with additional settings in the appendix, show that S-P is relatively robust and at least as good as random (often better).
>
>
>
> 7) ''Although the action space is state dependent $\mathcal A_s$ in the introduction ...’’
>
> The action space can indeed be state dependent; the bounds use the  product set for simplicity, but we will rewrite $|\mathcal S \times \mathcal A|$ as $\sum_s |\mathcal A_s|$, which is the more general version.
>
> 8) ''How does the bounds/results compare with a strategy that would be able to update a policy online?''
>
> This is an interesting issue. Our results show that for minimax bounds / worst-case settings, our algorithm matches (up to logs and constants) the best sample complexity results possible for the online setting and we will add in a short proof sketch of these sample complexity bounds to the appendix (see also our response to the first point raised by Reviewer mozb and to the first point raised by Reviewer `tDzP'). However empirically and in more structured  problems one can expect to learn faster with online algorithms. This is by design: an online algorithm can direct its effort towards the critical directions that help separate suboptimal arms from the optimal ones. More investigation on this tradeoff is important to understand whether creating an infrastructure for an online algorithm is worth the effort compared to an offline solution like ours. We will add a short discussion of this to the text.

---

> > ### Comment · Reviewer_UR5r · 2021-08-23
> > **Thank you**
> >
> > Thank you for the detailed answers to my comments that I read carefully.

---

### Decision · Program_Chairs · 2021-09-27

**Decision:**

Accept (Poster)

**Comment:**

This paper considers the following variant of the linear contextual bandit problem: Given historical data, design a non-reactive policy that can be used on future gather data that can be used to learn a near-optimal policy. The reviews felt that the problem is well-motivated, and that the problem setting and techniques both have some novelty. However, concerns were raised regarding the overlap between the algorithmic ideas used here and in previous work on reward-free exploration in RL; this should be addressed in the final version of the paper.